# REASONING PATH DIVERGENCE: A NEW METRIC AND CURATION STRATEGY TO UNLOCK LLM DIVERSE THINKING

## ABSTRACT

While Test-Time Scaling (TTS) effectively enhances the reasoning capabilities of Large Language Models (LLMs), its potential is often bottlenecked by low output diversity. This limitation raises questions about the standard *one problem, one solution* (1P1S) fine-tuning paradigm, which, by rewarding a single canonical answer, may encourage models to overfit to specific reasoning paths. To address this, we argue that adopting a *one problem, multiple solutions* (1PNS) training paradigm is crucial for cultivating reasoning diversity and unlocking the full potential of LLM reasoning. However, a central challenge of this paradigm lies in quantifying the semantic difference between complex, multi-step reasoning paths. To address this, we introduce Reasoning Path Divergence (RPD), a novel, fine-grained metric that operates at the step-level of Long Chain-of-Thought solutions. Using RPD, we curate a training set composed of maximally diverse solutions for each problem. Experiments with Qwen3-4B-Base demonstrate that training on our RPD-curated data significantly enhances output diversity and yields substantial gains in pass@k performance. Specifically, our 1PNS approach surpasses the 1P1S baseline by an average of 2.80% on pass@16 across challenging math benchmarks, with the improvement reaching 4.99% on AIME24, making Test-Time Scaling more effective.

## 1 INTRODUCTION

Large Language Models (LLMs) (Achiam et al., 2023; Chowdhery et al., 2023; Touvron et al., 2023) have achieved unprecedented success in complex reasoning domains, tackling challenges in areas like competitive mathematics and theoretical physics that were once considered beyond the reach of automated systems. This progress has been largely driven by Chain-of-Thought (CoT) prompting (Wei et al., 2022; Nye et al., 2021), which elicits step-by-step reasoning from language models. Building upon CoT, Test-Time Scaling (TTS) methods have become standard practice, particularly in recent models such as OpenAI's o1 series (Jaech et al., 2024). By generating multiple reasoning trajectories at inference time and selecting among them through techniques like Best-of-N sampling (Brown et al., 2024; Song et al., 2024) and self-consistency (Wang et al., 2022), TTS methods achieve substantial improvements on complex reasoning tasks. However, the effectiveness of TTS methods critically depends on the diversity of generated reasoning paths (Chen et al., 2025; Dang et al., 2025; Yao et al., 2025; Chow et al., 2025). When models produce only minor variations of the same flawed reasoning, the benefits of additional sampling diminish rapidly.

This diversity bottleneck arises in part from how current models are trained on reasoning tasks. Standard training datasets typically provide only a single solution path for each problem, teaching models to converge on one "correct" way of reasoning rather than exploring the space of valid alternatives. While prior work has proposed various modifications to loss functions to encourage diversity (Li et al., 2025c; Chen et al., 2025; Yao et al., 2025), fundamental questions about the relationship between training data diversity and model output diversity remain open. Therefore, the central question we explore in this paper is:

*Can a one problem, multiple solutions training paradigm effectively mitigate output homogenization and improve TTS performance?*

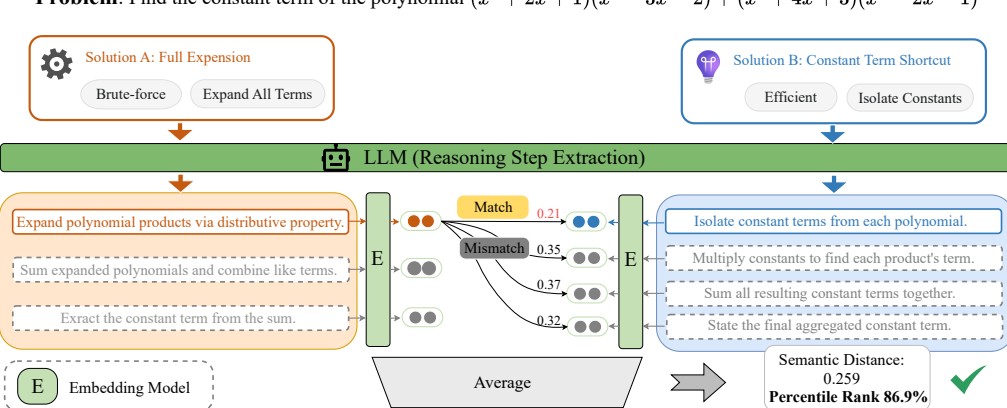

**Problem**: Find the constant term of the polynomial $(x^2 + 2x + 1)(x^2 - 3x - 2) + (x^2 + 4x + 3)(x^2 - 2x - 1)$

Figure 1: The workflow of our Reasoning Path Divergence (RPD) metric. Given two solutions (A and B), an LLM first decomposes them into step-level summaries. An asymmetric matching is then performed: each step in the shorter summary (A) is matched to its semantically closest counterpart in the longer summary (B) based on embedding cosine distance. The final RPD score is the average of these minimum distances. Detailed examples with analysis is provided in Appendix C.

In this work, we explore a pragmatic approach to address this diversity bottleneck: training models on datasets where each problem is paired with multiple distinct solutions. To construct such datasets, we first need to solve a fundamental challenge: reliably measuring semantic diversity between complex reasoning paths. Common approaches, such as computing cosine similarity on embeddings (Reimers & Gurevych, 2019) of the entire solution text, fail for Long Chain-of-Thought solutions because they conflate high-level strategic differences with low-level computational details and narrative style. To address this, we introduce Reasoning Path Divergence (RPD), a novel diversity metric that leverages Large Language Models to summarize solutions into their core logical steps, then employs an asymmetric matching process to quantify semantic overlap. This design enables RPD to distinguish genuine strategic novelty from superficial variations, providing the foundation for systematic diversity-driven data curation.

Equipped with this metric, we selected the OpenThought3 dataset (Guha et al., 2025) as our testbed. Its primary advantage is providing a large-scale collection of 53,125 mathematical problems, each accompanied by 16 Long-CoT answers. These properties establish the dataset as a premier testbed for our subsequent diversity-driven data curation experiments.

Our main contributions in this work are:

- **A Novel Metric and Diversity-Driven Curation Strategy.** We first propose and validate Reasoning Path Divergence (RPD), a novel metric for quantifying the semantic diversity between Long-CoT solutions. Building on this metric, we develop a data curation pipeline that systematically constructs a high-quality *one problem, multiple solutions* training set by selecting the most semantically distinct solutions for each problem.

- **Demonstrated Gains in Diversity and Performance.** Models fine-tuned on our multi-solution (1PNS) dataset achieve an average improvement of **2.80%** in $pass@16$ performance across challenging math benchmarks, highlighted by a peak gain of **4.99%** on the AIME24 benchmark, while simultaneously exhibiting higher output diversity as measured by our RPD metric. These gains demonstrate that multi-solution training effectively addresses the diversity bottleneck in Test-Time Scaling, thereby boosting its effectiveness.

## 2 RELATED WORK

**Test-Time Scaling.** A significant branch of Test-Time Scaling (TTS) focuses on improving performance by generating and aggregating multiple candidate solutions, which can be broadly divided into selection and fusion strategies. Selection-based methods identify the single best answer from

a pool of candidates, such as selecting the one with the highest verifier score in Best-of-N (Brown et al., 2024; Song et al., 2024) or the most frequent one via Majority Voting (Wang et al., 2022). To improve sample efficiency, some works filter candidates before the final selection or voting (Munkhbat et al., 2025; Chen et al., 2024; Wu et al., 2025). In contrast, fusion-based methods merge multiple answers, for instance, by prompting an LLM to act as a summarizer (Jiang et al., 2023; Li et al., 2025b;a). Crucially, the effectiveness of these methods is fundamentally bottle-necked by low output diversity, as conventional training encourages the model to overfit to a single, canonical reasoning path.

**LLM Generation Diversity.** A large body of work confirms that standard supervised fine-tuning is detrimental to generation diversity (O'Mahony et al., 2024; Chen et al., 2025; Li et al., 2025c), prompting explorations into various training-phase optimizations to mitigate this issue, especially as recent studies establish a strong positive correlation between a model's solution diversity and its reasoning potential (Yao et al., 2025). These algorithm-centric approaches are varied, ranging from modifying the training objective with techniques like confidence regularization (Chen et al., 2025) or direct Best-of-N optimization (Chow et al., 2025), to altering the training process via sparse updates (Li et al., 2025c), checkpoint ensembling (Dang et al., 2025), and lightweight, diversity-aware parameter tuning (Chung et al., 2025). Complementing these effective, algorithm-centric strategies, our work explores a data-centric perspective aimed at directly enriching the reasoning diversity within the training data itself.

**Data Curation.** The importance of curating high-quality and diverse datasets for fine-tuning is a widely recognized principle(Albalak et al., 2024). Existing efforts to enhance diversity have pri-marily targeted *inter-problem* diversity, focusing on ensuring a broad mix of distinct problems by using automated selection frameworks (Liu et al., 2024), removing semantic duplicates (Abbas et al., 2023), or optimizing domain mixtures (Xie et al., 2023). In contrast, cultivating *intra-problem* di-versity—teaching a model multiple ways to solve the same problem—remains a largely unexplored challenge, a critical gap that our work aims to fill.

## 3 METHOD

Enabling the *one problem, multiple solutions* training paradigm hinges on the ability to identify and select semantically distinct reasoning paths. To address this core challenge, this section first introduces Reasoning Path Divergence (RPD), a novel, fine-grained metric designed specifically for Long-CoT solutions. We then detail our 1PNS Curation Pipeline, which employs RPD to systemat-ically construct a high-diversity training set from the OpenThought3 dataset (Guha et al., 2025), a large collection of 53,125 mathematical problems, each accompanied by 16 Long-CoT answers.

### 3.1 REASONING PATH DIVERGENCE (RPD): A STEP-LEVEL DIVERSITY METRIC

Conventional metrics that apply embeddings to the full solution text are poorly-suited for assessing Long-CoT diversity. By flattening a solution's entire logical structure into a single vector, they conflate high-level strategic shifts with superficial textual variations. Our RPD metric overcomes this limitation by analyzing the reasoning process at the step-summary level, focusing on high-level logic rather than implementation details. The computation, illustrated in Figure 1, involves two core stages:

**1. Reasoning Step Extraction.** We use a LLM (Qwen3-14B; Team, 2025), guided by the prompt detailed in Appendix A.1, to decompose two Long-CoT solutions, $S_A$ and $S_B$, into their core logical steps. This process transforms each verbose solution into a structured list of concise step summaries: $L_A = \{a_1, ..., a_m\}$ and $L_B = \{b_1, ..., b_n\}$.

**2. Asymmetric Distance Computation.** The second stage quantifies the semantic distance between the two step lists using an asymmetric matching process. First, each step summary is converted into a high-dimensional vector using Qwen3-Embedding-8B (Zhang et al., 2025). Next, we identify the solution with fewer steps ($m \leq n$) as the reference, say $S_A$, and for each of its steps $a_i$, we find its closest semantic match within the other solution, $S_B$, by calculating the minimum cosine distance:

$$d_i = \min_{j=1,...,n} \left( 1 - \frac{\vec{e}_{a_i} \cdot \vec{e}_{b_j}}{\|\vec{e}_{a_i}\| \|\vec{e}_{b_j}\|} \right) \tag{1}$$

The overall RPD score, $D(S_A, S_B)$, is the average of these minimum distances:

$$D(S_A, S_B) = \frac{1}{m} \sum_{i=1}^{m} d_i \tag{2}$$

The robustness of this asymmetric design stems from its ability to handle potential inconsistencies in summarization granularity. It quantifies how well the core logic of the shorter path is covered by the longer one. This ensures that if one solution is simply a more detailed variant of another, the RPD score will be low, whereas fundamentally different strategies will yield a high score. The complete procedure is formalized in Appendix A.2.

### 3.2 THE 1PNS CURATION PIPELINE

Our pipeline curates the raw OpenThought3 dataset into a high-diversity 1PNS training set through two main phases.

**Phase 1: Initial Quality Filtering.** We began with a pool of 10,000 mathematical problems from OpenThought3. Given the absence of ground-truth labels, we first applied a multi-stage filtering protocol to ensure data quality. The protocol involved two key steps: first, length-based filtering to help determine a practical `max_new_tokens` for inference, and second, an LLM-based screening (using Qwen3-14B) to discard ambiguous problems and solutions that were incomplete or lacked a final answer. This initial phase yielded a high-quality candidate set of 1,600 problems, each with at least 10 candidate solutions that passed the screening protocol. The specifics of this protocol are detailed in Appendix B.1.

Before proceeding to the core selection, we investigated the natural diversity within this candidate set using a summary-based LLM Judge. As detailed in Appendix B.2, we prompted a Qwen3-14B model to assess the overall diversity across the set of all candidate solution summaries for each problem. The analysis showed a significant lack of diversity: a majority of problems, **58%**, were found to contain solutions that all followed the same single reasoning strategy, with only minor variations. This observation underscores that a high number of solutions does not inherently guarantee strategic reasoning diversity, making an explicit problem selection phase essential.

**Phase 2: Diversity-Driven Selection.** This phase consists of a two-stage process guided by our RPD metric:

**1. Problem Selection.** We first rank problems by their intrinsic solution diversity. For each problem $P$ with $k$ solutions, we compute its overall diversity score, $\text{Score}_{\text{div}}(P)$, by averaging the pairwise RPD scores across all its unique solution pairs:

$$\text{Score}_{\text{div}}(P) = \frac{2}{k(k-1)} \sum_{1 \le i < j \le k} D(S_i, S_j)$$

We then select the top-$N$ problems from this ranked list.

**2. Solution Selection.** For each of the top-$N$ problems, we then curate a concise set of $M$ maximally diverse solutions. This is accomplished using a greedy algorithm that iteratively selects the solution exhibiting the highest average RPD to the already chosen subset.

This two-stage process results in a final training set rich in strategically diverse reasoning paths. The detailed algorithm is provided in Appendix A.3.

## 4 EXPERIMENTS

To validate the core hypothesis of our work—that diversity-driven data curation can enhance a model's Test-Time Scaling (TTS) performance, we designed and conducted a series of experiments. Our evaluation is twofold: first, we directly assess how effectively our proposed Reasoning Path Divergence (RPD) metric identifies strategic diversity among solutions; second, we evaluate the impact of a training set curated with this metric on the final `pass@k` performance of a model in the downstream reasoning task.

## 4.1 RPD METRIC EVALUATION

**Setup.** To evaluate RPD's effectiveness in identifying semantically diverse reasoning paths, we randomly sample 100 problems and their solutions from the high-quality candidate set established in our curation pipeline (Sec. 3.2). For each problem, every compared method selects the pair of solutions it predicts to be the most diverse. A powerful LLM Judge then assesses whether the selected pair exhibits diverse problem-solving approaches and strategies, and we report the **success rate** as our primary evaluation criterion. The reliability of this LLM Judge has been validated against human annotations (see Appendix D.1.2 for the full prompt and alignment study).

**Methods Compared.** We evaluate the following methods:

- **Random**: Randomly selects a pair of solutions, serving as a lower-bound baseline.
- **Raw Embedding (Raw Emb.)**: Selects the pair with the greatest cosine distance between the embeddings of the full solution texts.
- **Summary Embedding (Summary Emb.)**: Selects the pair with the greatest cosine distance between the embeddings of solution summaries.
- **LLM Selection**: A LLM (Qwen3-14B) selects the most diverse pair based on the summaries of all candidate solutions (see Appendix D.1.1 for details).
- **Ours (RPD)**: Our proposed asymmetric, step-level semantic distance metric.

**Results and Analysis.** As shown in Table 1, our RPD metric achieves a **53%** success rate, significantly outperforming all baselines, including those based on raw embeddings (40%), summary embeddings (48%), and even a powerful LLM selector (44%). These results offer two key insights. First, RPD's fine-grained, step-level analysis is crucial for overcoming the limitations of holistic embedding methods that conflate high-level strategy with superficial text. Second, its systematic pairwise comparison proves more robust than a heuristic LLM judgment when faced with identifying the most diverse pair from a large candidate pool. This performance confirms RPD's effectiveness as an automated metric for our diversity-driven curation pipeline.

Table 1: Effectiveness of various diversity metrics.

| Method | Success Rate (%) |
|---|---|
| Random | 27 |
| Raw Emb. | 40 |
| LLM Selection | 44 |
| Summary Emb. | 48 |
| **Ours (RPD)** | **53** |

## 4.2 EFFECTIVENESS OF MULTI-SOLUTION FINE-TUNING

In this experimental section, we aim to answer the following research questions:

**Q1:** Does fine-tuning with the *one problem, multiple solutions* (1PNS) paradigm lead to superior downstream reasoning performance, as measured by `pass@k`, compared to the standard *one problem, one solution* (1P1S) approach?

**Q2:** Within the 1PNS paradigm, does curating solutions for high strategic diversity using our RPD metric yield better `pass@k` performance than other baselines?

### 4.2.1 EXPERIMENTAL SETUP

**Model.** We use the Qwen3-4B-Base model (Team, 2025) for our primary experiments. To ensure the robustness of our findings, corresponding results for the Qwen2.5-3B model (Team, 2024) are provided in the Appendix E.2.

**Benchmark.** We evaluate the model's performance on three challenging mathematical reasoning benchmarks that align with our training data domain: AIME24[1], MATH500 Level 5 (Hendrycks et al., 2021), and Olympiad Bench[2] (He et al., 2024). Performance is measured using the `pass@k` metric.

---

[1] https://huggingface.co/datasets/Maxwell-Jia/AIME_2024

[2] For our evaluation, we selected an English, text-only, deterministic-answer mathematical subset of the Olympiad Bench to align with our training set.

**Baselines.** To comprehensively evaluate our diversity-driven data curation method, we conduct a comparison against several baselines. For our main experiments, we standardize the multi-solution format to **one problem and three solutions (1P3S)**. The impact of varying the number of solutions per problem is investigated in our ablation studies (Sec 4.2.3). To ensure a fair comparison, the total number of training instances is held constant at 300 across all methods.

Our proposed method, **Ours (RPD)**, constructs a training set of 100 problems and 3 solutions per problem, guided by our RPD metric's diversity scores. We compare it against the following baselines, which are grouped into two categories. The detailed construction methodology for each is provided in Appendix D.2.

*1. Comparison of 1P1S vs. 1P3S paradigms.*

- **Random 1P1S:** The standard SFT baseline, constructed by randomly selecting 300 unique problems and pairing each with one randomly chosen solution. This baseline is used to measure the fundamental performance gain of the 1P3S approach.

*2. Comparison of diversity selection metrics (all using a 1P3S structure).*

- **Random 1P3S:** A naive multi-solution approach. We randomly select 100 problems and use 3 randomly chosen solutions for each.
- **LLM Selection:** An LLM is prompted to select 100 problems and generate 3 diverse solutions for each.
- **Raw Embbeding (Raw Emb.) :** We select the 100 problems and 3 corresponding solutions that maximize diversity based on the cosine distance between the embeddings of the full answer texts.
- **Summary Embbeding (Summary Emb.):** We select data by maximizing the cosine distance between embeddings of AI-generated answer summaries for 100 problems and their 3 solutions.

**Implementation Details.** We fine-tune the Qwen3-4B-Base model using supervised fine-tuning with 4-bit QLoRA (rank=16, alpha=32). The model is trained for 12 epochs in BF16 precision on NVIDIA H20 GPUs. We use the AdamW optimizer with an batch size of 16 and a cosine learning rate scheduler, peaking at $5 \times 10^{-5}$. For inference, we use nucleus sampling (temperature=0.6, top_p=0.95) with maximum generation lengths tailored to each benchmark (14K for AIME24, 10K for MATH500, 8K for Olympiad). To ensure statistical robustness, we report average scores over multiple runs (4 for AIME24/MATH500, 2 for Olympiad).

### 4.2.2 RESULTS AND ANALYSIS

To answer our research questions, we present the experimental results in two parts. First, we compare the *one problem, multiple solutions* (1PNS) paradigm against the standard *one problem, one solution* (1P1S) baseline. Second, we evaluate the effectiveness of our diversity metric against various alternative selection strategies.

**Q1: Superiority of the 1PNS Paradigm**

To address our first research question, we compare the performance of our 1P3S training method against the standard 1P1S baseline across all three benchmarks.

As shown in Figure 2, while the performance of our `1PNS` approach is comparable to the `1P1S` baseline at `pass@1`, it significantly outperforms the baseline at larger values of $k$. On average, our method achieves a `pass@16` gain of **2.80%** across all benchmarks. The improvement peaks on highly challenging mathematical reasoning problems like AIME24, with the gain reaching **4.99%** on this benchmark. These results support our core hypothesis that the *one problem, multiple solutions* paradigm is a more effective strategy for enhancing the Test-Time Scaling performance of models on complex reasoning tasks.

**Q2: Effectiveness of the RPD Metric**

Next, we evaluate how our diversity metric performs against other data selection strategies. All compared selection strategies follow the *1P3S (one problem, three solutions) format*. Table 2 presents

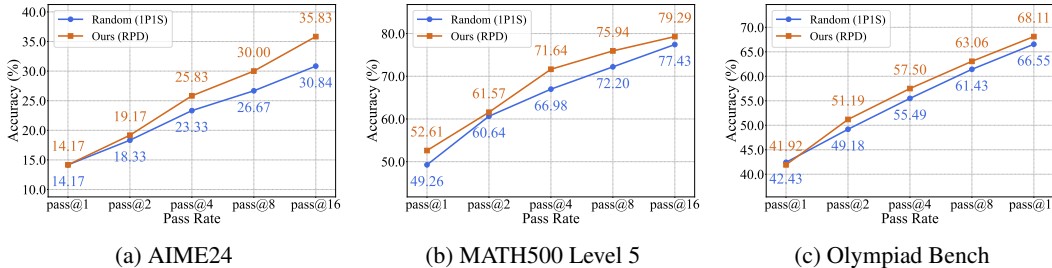

(a) AIME24      (b) MATH500 Level 5      (c) Olympiad Bench

Figure 2: Performance comparison of our 1P3S approach against the 1P1S baseline across three mathematical reasoning benchmarks. Each subplot corresponds to a different benchmark, showing the pass@k accuracy for k=1, 2, 4, 8, 16.

Table 2: Comparison of different diversity selection methods on the MATH500 Level 5 benchmark. All methods except *Base* use a **1P3S** (100 problems, 3 solutions) structure.

| Method | pass@1 (%) | pass@2 | pass@4 | pass@8 | pass@16 |
|---|---|---|---|---|---|
| *Base* | 46.08 | 56.90 | 64.37 | 71.27 | 75.00 |
| Random (1P3S) | 49.07 | 59.70 | 68.66 | 73.32 | 77.24 |
| Raw Emb. | 50.19 | 59.14 | 67.54 | 71.64 | 77.80 |
| Summary Emb. | 52.24 | 59.89 | 68.66 | 73.14 | 77.43 |
| LLM Selection | 49.81 | 58.96 | 66.23 | 73.51 | 77.61 |
| **Ours (RPD)** | **52.61** | **61.57** | **71.64** | **75.94** | **79.29** |

the results on the MATH500 Level 5 benchmark (results for AIME24 and Olympiad Bench are in Appendix E.1).

The results in Table 2 demonstrate that our RPD-guided data selection method consistently outperforms all baseline strategies across every `pass@k` metric. While some methods, such as Summary Emb., are competitive at `pass@1`, our approach establishes a more decisive lead at higher values of $k$. For instance, it creates a nearly **+3.0%** performance gap over the next-best strategies at `pass@4`. This performance gap highlights RPD's superior ability to discern true strategic diversity, a quality not fully captured by holistic embedding distances or heuristic LLM selection.

### 4.2.3 ABLATION STUDIES

We conduct a series of ablation studies to provide a comprehensive analysis of our method and its properties. We begin by evaluating our method's impact on the diversity of generated solutions. We then analyze the sensitivity to a key hyperparameter—the number of solutions curated for each problem—before disentangling the individual contributions of our problem and answer selection components. Subsequently, we explore the interplay between our fine-tuning approach and inference-time temperature sampling. Finally, we validate the scalability of our paradigm on a larger training set.

**Analysis of Solution Diversity.** To verify that 1PNS training increases output diversity, we analyzed 16 generated solutions for each problem in MATH Level5 test set. We partitioned problems into a *moderately-solved group* (2-12 correct solutions) and a *well-solved group* (13-16 correct solutions) to analyze performance on problems of varying difficulty. Diversity was measured using our RPD metric, which is the average pairwise RPD among correct solutions within each problem, and Div-Self-BLEU (100 - Self-BLEU) (Kirk et al., 2023). For both metrics, a higher score indicates greater output diversity.

The results in Table 3 reveal a notable adaptability in our fine-tuned model. On moderately-solved (i.e., more difficult) problems, it generates the most diverse outputs, as measured by both RPD and Div-Self-BLEU. On well-solved problems, however, its output diversity is lower than the 1P1S baseline, indicating a high-confidence convergence. We interpret this behavior as a highly efficient strategy for Test-Time Scaling: the model learns to *selectively apply exploration on challenging problems while defaulting to exploitation on simpler ones*. This adaptability is key to optimizing overall `pass@k` performance.

Table 3: Diversity scores for different methods on the MATH500 Level 5 test set, evaluated on the Qwen3 4B Base model. Scores are partitioned by the number of correct solutions (pass count) out of 16 attempts.

| Method | Div-Self-BLEU | | Our Metric | |
|---|---|---|---|---|
| | Pass Count 2-12 | Pass Count 13-16 | Pass Count 2-12 | Pass Count 13-16 |
| Random (1P1S) | 35.27 | 15.26 | 15.17 | 13.30 |
| Random (1P3S) | 32.52 | 14.62 | 15.57 | 14.00 |
| LLM Selection ((1P3S)) | 36.36 | 14.19 | 15.11 | 13.31 |
| Raw Emb. (1P3S) | 33.94 | 14.23 | 15.39 | 13.08 |
| Summary Emb. (1P3S) | 37.42 | 14.46 | 15.69 | 12.98 |
| **RPD (1P3S)** | 38.20 | 14.31 | 15.80 | 12.62 |

Table 4: Ablation study on the number of diverse solutions selected by **our RPD metric** per problem on the MATH500 Level 5 benchmark, compared against the 1P1S baseline. The total sample size is kept constant at 300.

| Configuration | pass@1 (%) | pass@2 | pass@4 | pass@8 | pass@16 |
|---|---|---|---|---|---|
| Random (1P1S) | 49.26 | 60.64 | 66.98 | 72.20 | 77.43 |
| RPD (1P2S) | 52.43 | **61.57** | 69.96 | 74.63 | 77.99 |
| **RPD (1P3S)** | 52.61 | **61.57** | **71.64** | **75.94** | **79.29** |
| RPD (1P4S) | 52.24 | 59.70 | 70.90 | 74.63 | 79.10 |
| RPD (1P5S) | **53.92** | 61.20 | 67.73 | 73.88 | 78.54 |

Table 5: Ablation study on the contributions of the problem (Q) and answer (A) selection components on the MATH500 Level 5 benchmark. All configurations use a 100Q, 3A structure.

| Method (Problem + Answer) | pass@1 (%) | pass@2 | pass@4 | pass@8 | pass@16 |
|---|---|---|---|---|---|
| Random-Q + Random-A | 49.07 | 59.70 | 68.66 | 73.32 | 77.24 |
| Random-Q + RPD-A | 50.93 | 61.38 | 68.47 | 73.69 | 77.43 |
| RPD-Q + Random-A | 49.81 | 59.52 | 67.91 | 74.82 | 78.36 |
| **RPD-Q + RPD-A (Ours)** | **52.61** | **61.57** | **71.64** | **75.94** | **79.29** |

**Impact of the Number of Solutions per Problem.** Next, we investigate how the number of solutions for each problem affects final model performance, keeping the total training sample size fixed at 300. As shown in Table 4, we compare our method's performance when configured to select two, three, four, and five diverse solutions per problem against the standard single-solution baseline.

The results in Table 4 first and foremost demonstrate the clear superiority of the 1PNS paradigm, as all multi-solution configurations consistently outperform the single-solution baseline, particularly at larger values of k. Focusing on these metrics reveals an optimal balance: performance peaks with the *RPD (1P3S)* configuration and declines as more solutions are added per problem. This suggests a critical trade-off between "diversity depth" and "problem breadth," and while the optimal balance is likely contingent on the source data's intrinsic diversity, the 1P3S configuration proves to be the most effective for the OpenThought3 dataset.

**Quantifying the Impact of Problem and Answer Selection Strategies.** To disentangle the individual contributions of our problem selection (`RPD-Q`) and answer selection (`RPD-A`) strategies, we evaluate our full method against ablations where each component is replaced by a random selection baseline. The results are presented in Table 5.

The results in Table 5 lead to three key findings. First, any configuration incorporating our diversity-driven selection—either for problems or answers—outperforms the fully random baseline at larger `pass@k` values. Second, when comparing their individual impacts, problem selection (`RPD-Q`) is more critical for enhancing Test-Time Scaling, providing a **+1.12%** gain at `pass@16` over the random baseline, substantially larger than the **+0.19%** gain from selecting for diverse answers (`RPD-A`) alone. Finally, our full method, which combines both strategies, achieves the best performance by a

Table 6: Performance comparison on the MATH500 Level 5 benchmark between our method and the random baseline across various sampling temperatures ($T$).

| Method | Temp ($T$) | pass@1 (%) | pass@2 | pass@4 | pass@8 | pass@16 |
|--------|-----------|-----------|--------|--------|--------|---------|
| Random | 0.2 | **51.12** | **58.96** | 65.30 | 69.96 | 73.13 |
| **RPD** | 0.2 | 50.00 | 57.46 | **66.79** | **71.46** | **74.82** |
| Random | 0.4 | **54.11** | **60.26** | 68.10 | 72.39 | 76.31 |
| **RPD** | 0.4 | 47.95 | 60.08 | **69.03** | **75.19** | **77.80** |
| Random | 0.6 | 49.26 | 60.64 | 66.98 | 72.20 | 77.43 |
| **RPD** | 0.6 | **52.61** | **61.57** | **71.64** | **75.94** | **79.29** |
| Random | 0.8 | **51.87** | **61.57** | 69.22 | 74.44 | 78.36 |
| **RPD** | 0.8 | 50.56 | 60.82 | **69.47** | **74.82** | **78.92** |
| Random | 1.0 | 45.34 | 59.71 | 68.66 | **73.88** | 76.87 |
| **RPD** | 1.0 | **48.51** | **59.89** | **69.22** | **73.88** | **77.80** |

significant margin (e.g., improving `pass@16` by nearly another full percentage point over the next-best configuration). This demonstrates a clear synergistic effect, confirming that while `Diverse-Q` provides a strong foundation, both components are indispensable for maximizing reasoning performance.

**Interaction with Inference-Time Sampling Temperature.** A common method for increasing output diversity at inference time is to raise the sampling temperature ($T$). A key question is whether the diversity benefits from our fine-tuning method are redundant with, or complementary to, this technique. To investigate this, we evaluate the performance of our method (RPD 1P3S) against the baseline (Random 1P1S) under five different temperature settings, from low ($T = 0.2$) to high ($T = 1.0$). We set `top_p` to 0.95 and `top_k` to -1.

The results presented in The results in Table 6 show that the performance gap between our method and the baseline widens as $k$ increases, regardless of temperature. For smaller values of $k$ (e.g., `pass@1`, `pass@2`), the performance between our method and the random baseline is competitive, with neither showing a decisive advantage across all temperatures. However, as $k$ increases, a clear and consistent pattern emerges: at larger values of $k$ (`pass@8` and `pass@16`), our method consistently outperforms the baseline across the entire spectrum of temperature settings.

This observation directly supports our conclusion: our RPD-guided training is orthogonal and complementary to inference-time temperature sampling. Our method fundamentally enriches the model's accessible solution space by exposing it to diverse reasoning pathways during training. Temperature, in contrast, acts as an independent tool to control the stochasticity of navigating that solution space at inference time. The consistent performance advantage at higher $k$-values confirms that our approach provides a distinct and foundational benefit that is not made redundant by simply tuning inference-time parameters.

**Scalability to Larger Datasets.** We further validated the scalability of our 1PNS paradigm by increasing the training set size to 1,500 samples. As detailed in the Appendix E.3, our diversity-driven strategy maintains its significant performance advantage over the traditional 1P1S baseline at this larger scale.

## 5 CONCLUSION

To enable the *one problem, multiple solutions* (1PNS) paradigm, we introduce a novel metric for quantifying reasoning diversity, Reasoning Path Divergence (RPD), and leverage it to curate a dataset of maximally diverse solutions. Our experiments validate the superiority of the 1PNS paradigm over the standard 1P1S baseline, as training on this RPD-curated data mitigates output homogenization while yielding significant `pass@k` gains. These findings establish that our proposed approach provides a direct pathway to boosting the effectiveness of test-time scaling.

## ETHICS STATEMENT

The research presented in this paper adheres to the ICLR Code of Ethics. Our work is motivated by the goal of advancing machine learning research and we have carefully considered its potential ethical implications. All datasets and models utilized in our experiments are publicly available and open-source, and we have followed all their terms of use. We acknowledge that our methods could have unforeseen applications, and we encourage the community to build upon our work with a strong consideration for societal impact and fairness. To the best of our knowledge, our work does not introduce new biases, and we have been transparent in our experimental setup and reporting to allow for community scrutiny.

## REPRODUCIBILITY STATEMENT

We are committed to the full reproducibility of our research. To facilitate this, we provide detailed descriptions of our algorithms, model architectures, and key hyperparameters within the main paper and a comprehensive appendix. Crucially, as our methodology involves large language models, we have included the exact prompts used for our experiments in the appendix to ensure transparency and replicability. Our entire experimental framework relies exclusively on publicly available open-source models, standard benchmarks, and datasets, removing barriers to independent verification. We believe the extensive details provided in our paper and appendices are sufficient for our peers to reproduce our core results with ease.

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

## THE USE OF LARGE LANGUAGE MODELS (LLMS)

In accordance with the ICLR 2026 policy, we disclose that a large language model (LLM) was used as a writing-assistance tool in the preparation of this manuscript. Its role was strictly limited to minor copy-editing tasks, such as improving grammar, rephrasing sentences for clarity, and polishing the overall language. The LLM did not contribute to any of the core research ideas, methodologies, experimental designs, or result interpretations presented herein. The authors have meticulously reviewed all text and take full responsibility for the scientific integrity and accuracy of the entire paper's content.

## A  RPD CURATION METHOD IMPLEMENTATION

### A.1  STEP-WISE SOLUTION SUMMARIZATION VIA LLM

Our proposed diversity metric relies on a fine-grained, step-by-step summary of the reasoning path for each solution. To create these summaries, we use an LLM (Qwen3-14B) to break down each solution into its core logical steps. A key challenge is to ensure these summaries accurately reflect the original methodology while maintaining a consistent level of granularity. Overly concrete summaries might capture superficial numerical differences, while overly abstract summaries might fail to distinguish between genuinely different strategies.

To solve this, we design a detailed prompt that controls the LLM's output format and level of abstraction. This prompt instructs the model to produce a structured JSON object containing 3 to 5 method-focused steps. This strict format helps maintain uniformity across all summarized solutions. The complete prompt is provided below.

---

**Prompt for Step-wise Solution Summarization**

You are a specialized AI expert in analyzing mathematical solutions. Your task is to first provide a step-by-step analysis of a solution, and then, based on your analysis, generate a final JSON output that is concise, direct, and method-focused.

REQUIRED OUTPUT STRUCTURE

Your response **MUST** have two distinct parts in the following order:

**Part 1: Analysis & Thinking Process**

- Start this section with the heading `### Analysis`.
- Briefly explain your reasoning as you deconstruct the provided solution. This is your "scratchpad".

**Part 2: Final JSON Output**

- After your analysis, provide the final JSON output enclosed in `//boxed{{}}`.
- This part must contain *only* the `//boxed{{...}}` block and nothing else.

CONTENT RULES FOR THE FINAL JSON

1. **Step Count**: The JSON must contain **strictly 3 to 5 logical steps**.
2. **Output Style**:
    - **Use direct, active verb phrases.** Start each description with a verb (e.g., "Calculate", "Identify", "Apply").
    - **DO NOT use narrative phrasing** like "The author identifies..." or "The solution then calculates...".
3. **Abstraction Level**:
    - Be abstract about numbers and variables, but **be specific about the methodology**.
    - **BAD (Too Vague):** "Use a formula to get the result."
    - **BAD (Too Concrete):** "Calculate 1/3 + 1/6 = 1/2."
    - **GOOD (Balanced):** "Combine the individual rates to find the total work rate."

---

JSON STRUCTURE SPECIFICATION
- The root object must have one key: `logical_steps`.
- The value of `logical_steps` must be a list (`[]`) of step objects.
- Each step object (`{{}}`) must contain two keys:
  - `step_title`: A short title for the step (e.g., "Step 1: Combine Rates"). Use `null` if not applicable.
  - `step_description`: A concise summary of the action, following all rules above.

EXAMPLE OF THE COMPLETE TWO-PART OUTPUT

**Input Solution**: "Pipe A fills a tank in 3 hours, so its rate is 1/3 tank/hr. Pipe B fills it in 6 hours, so its rate is 1/6 tank/hr. Together, their rate is 1/3 + 1/6 = 1/2 tank/hr. Therefore, the time to fill the tank together is the reciprocal of the rate, which is 1 / (1/2) = 2 hours."

**Your Required Output**:

```
### Analysis
The solution addresses a classic work-rate problem.
1.  First, it calculates the individual rate for each pipe.
2.  Second, it sums these rates to get a combined rate.
3.  Finally, it converts the combined rate back into total
    time.
The logic is broken down into three clear, abstract steps.

//boxed{{
  "logical_steps": [
    {{
      "step_title": "Step 1: Determine Individual Rates",
      "step_description": "Determine the individual work rate
          of each component based on the time taken."
    }},
    {{
      "step_title": "Step 2: Combine Rates",
      "step_description": "Combine the individual rates to
          find the total system work rate."
    }},
    {{
      "step_title": "Step 3: Calculate Total Time",
      "step_description": "Calculate the total time by taking
          the reciprocal of the combined work rate."
    }}
  ]
}}
```

---

YOUR TASK

**Math Problem**:
{question_text}

**Chain-of-Thought Solution to Analyze**:
{answer_cot}

## A.2 REASONING PATH DIVERGENCE (RPD) CALCULATION

After summarizing each solution into a series of core logical steps, the next phase is to compute the pairwise diversity using our **Reasoning Path Divergence (RPD)** metric. RPD is designed to quantify the semantic distance between the step-lists of two solutions, $S_A$ and $S_B$.

The calculation begins by embedding each logical step using the **Qwen3-Embedding-8B** model. Subsequently, it computes an asymmetric score by finding the average minimum cosine distance from the steps of the shorter solution to all steps in the longer one. This asymmetric design is crucial: it ensures that a solution containing a genuinely novel step is considered distant, even if its other steps are subsumed by a more comprehensive solution. The formal algorithm is detailed below.

---

**Algorithm 1** Reasoning Path Divergence (RPD) Calculation

---

**Require:** Two Long-CoT solutions, $S_A$ and $S_B$.
**Ensure:** A scalar diversity score $D \in [0, 1]$.
 1: $L_A \leftarrow \text{ExtractSteps}(S_A)$; $L_B \leftarrow \text{ExtractSteps}(S_B)$
 2: **if** $L_A$ is empty or $L_B$ is empty **then**
 3:     **return** $1.0$
 4: **end if**

 5: $E_A \leftarrow \{\text{Embed}(a_i) \mid a_i \in L_A\}$; $E_B \leftarrow \{\text{Embed}(b_j) \mid b_j \in L_B\}$

 6: $(E_{\text{shorter}}, E_{\text{longer}}) \leftarrow \begin{cases} (E_A, E_B) & \text{if } |E_A| \leq |E_B| \\ (E_B, E_A) & \text{otherwise} \end{cases}$

 7: $\text{min\_distances} \leftarrow \emptyset$
 8: **for all** $\vec{e}_s \in E_{\text{shorter}}$ **do**
 9:     $d_{\min} \leftarrow \min_{\vec{e}_l \in E_{\text{longer}}} \left(1 - \frac{\vec{e}_s \cdot \vec{e}_l}{\|\vec{e}_s\| \|\vec{e}_l\|}\right)$
10:     $\text{min\_distances} \leftarrow \text{min\_distances} \cup \{d_{\min}\}$
11: **end for**

12: $D_{\text{final}} \leftarrow \text{Mean}(\text{min\_distances})$
13: **return** $D_{\text{final}}$

---

## A.3 DIVERSITY-DRIVEN DATA CURATION

Our data curation process is a two-stage procedure designed to build a training set rich in strategic diversity. First, we perform **Problem Selection** to identify problems that naturally exhibit a wide range of solutions by scoring each problem based on its total intrinsic diversity. Second, for each of these top-ranked problems, we execute a greedy **Solution Selection** algorithm to curate a small but maximally diverse subset of $M$ solutions. This two-stage approach ensures both inter-problem and intra-problem diversity. The algorithms for both stages are detailed below.

---

**Algorithm 2** Stage 1: Problem selection by intrinsic diversity

---

**Require:** Candidate problem set $\mathcal{P}$, target count $N$, pairwise distance function $D(\cdot, \cdot)$
**Ensure:** Top-$N$ problems $\mathcal{P}_{\text{top}}$ ranked by intrinsic diversity
 1: Initialize empty list of pairs $\mathcal{L} \leftarrow []$
 2: **for all** problem $P \in \mathcal{P}$ **do**
 3:     Let $\mathcal{S}_P = \{S_1, \ldots, S_{k_P}\}$ be its candidate solutions
 4:     **if** $k_P < 2$ **then**
 5:         append $(P, -\infty)$ to $\mathcal{L}$
 6:         **continue**
 7:     **end if**
 8:     Compute all pairwise distances $\{D(S_i, S_j) : 1 \le i < j \le k_P\}$
 9:     $\text{avgD} \leftarrow \dfrac{2}{k_P(k_P - 1)} \sum_{i<j} D(S_i, S_j)$
10:     append $(P, \ \text{avgD})$ to $\mathcal{L}$
11: **end for**
12: Sort $\mathcal{L}$ by score (second element) in descending order
13: $\mathcal{P}_{\text{top}} \leftarrow$ first $\min(N, |\mathcal{P}|)$ problems from sorted $\mathcal{L}$
14: **return** $\mathcal{P}_{\text{top}}$

---

**Algorithm 3** Stage 2: Greedy Selection

---

**Require:** Candidate solutions $\mathcal{S}_{\text{cand}} = \{S_1, \ldots, S_k\}$, pairwise distance matrix $\mathbf{D} \in \mathbb{R}^{k \times k}$, target size $M$
**Ensure:** Selected index set $\mathcal{I}_{\text{select}}$ with $|\mathcal{I}_{\text{select}}| = \min(M, k)$
 1: **if** $M \le 0$ or $k = 0$ **then return** $\emptyset$
 2: **end if**
 3: **if** $M \ge k$ **then return** $\{1, \ldots, k\}$
 4: **end if**
 5: $i_{\text{first}} \leftarrow \arg\max_i \sum_{j \ne i} \mathbf{D}_{ij}$
 6: $\mathcal{I}_{\text{select}} \leftarrow \{i_{\text{first}}\}$;   $\mathcal{I}_{\text{remain}} \leftarrow \{1, \ldots, k\} \setminus \{i_{\text{first}}\}$
 7: **for each** $r \in \mathcal{I}_{\text{remain}}$ set $m[r] \leftarrow \mathbf{D}_{r, i_{\text{first}}}$
 8: **while** $|\mathcal{I}_{\text{select}}| < M$ and $\mathcal{I}_{\text{remain}} \ne \emptyset$ **do**
 9:     $r^\star \leftarrow \arg\max_{r \in \mathcal{I}_{\text{remain}}} m[r]$
10:     $\mathcal{I}_{\text{select}}.\text{append}(r^\star)$;   $\mathcal{I}_{\text{remain}}.\text{remove}(r^\star)$
11:     **for each** $r \in \mathcal{I}_{\text{remain}}$: $m[r] \leftarrow \min\left(m[r], \mathbf{D}_{r, r^\star}\right)$
12: **end while**
13: **return** $\mathcal{I}_{\text{select}}$

---

# B    DATASET PREPROCESSING AND ANALYSIS

## B.1    DETAILED DATASET FILTERING PROTOCOL

The OpenThought3 dataset is a valuable open-source resource, containing approximately 53,000 mathematical problems, each with 16 corresponding completions. However, the raw dataset presents several challenges for direct use in supervised fine-tuning. Key issues include the absence of ground truth labels, the possibility of encountering ambiguous or ill-posed problems, and the fact that some solutions may be unfinished or lack a definitive final answer. Furthermore, the length of the provided solutions varies dramatically.

To curate a high-quality training corpus and ensure computational efficiency during model infer-ence, we implement a rigorous two-stage filtering protocol on a subset of 10,000 problems from OpenThought3. This protocol addresses both solution length and quality.

**Stage 1: Length-Based Filtering.**    Our first step is to control for solution length. This measure is primarily motivated by the practical need to set a reasonable `max_new_tokens` parameter during inference. Accordingly, we filter out any problem whose average token count across all its solutions exceeds 14,000 tokens.

**Stage 2: Quality and Completeness Filtering.**    Next, we address the issue of solution quality and completeness. We employ an LLM (Qwen3-14B) as a judge to verify whether each solution is valid. For every solution in the length-filtered set, we provide its final 500 tokens as input to the LLM. The model is instructed to determine if the solution concludes properly by presenting a clear and final answer. Solutions that the LLM judge flags as incomplete or inconclusive are discarded, and any problem subsequently left with fewer than 10 valid solutions is also removed.

This comprehensive filtering pipeline refines the initial pool of 10,000 problems into a high-quality, curated set of **approximately 1,600 problems**. Each problem in this final set has an average solution length of less than 14,000 tokens and is accompanied by at least 10 complete, validated solutions. This curated 1,600-problem dataset serves as the foundation for all subsequent experiments con-ducted in this work.

## B.2    DATASET DIVERSITY ANALYSIS

To better inform our data curation, we first analyze the existing strategic diversity within our high-quality candidate set. We use a summary-based LLM Judge to classify whether the solutions for each problem are strategically uniform or diverse.

For each problem, we concatenate the step-wise summaries of all its candidate solutions (detailed in Appendix A.1) into a single string. This, along with the original problem statement, is provided to an LLM Judge (Qwen3-14B). The judge's task is to perform a binary classification on the entire set of solutions, identifying if at least two different solution strategies are present.

We specifically write the prompt to instruct the model to ignore superficial differences in wording or calculation, and instead focus on fundamental strategic choices, such as using direct casework ver-sus complementary counting. We do this so that the classification reflects genuine methodological diversity, not just surface-level variations. The insights from this analysis, as reported in the main text, confirm the need for our subsequent diversity-driven problem selection phase. The complete prompt for this task is detailed below.

> **Prompt for Problem Classification**
>
> You are a master mathematician and an expert in pedagogical analysis.  Your task is to classify a problem based on the methodological diversity of its proposed solutions.
> Your goal is to perform a binary classification:
>
> - **Class 2 (Diverse):** If there are at least two distinct core methodologies present across all the provided solution summaries.

- **Class 1 (Not Diverse):** If all solutions use the same core methodology, or if the differences are only superficial (e.g., a different order of calculation, or using standard procedural equivalents like substitution vs. elimination).

---

1. YOUR ANALYSIS FRAMEWORK & CORE CRITERIA

---

Your primary task is to act as a discerning analyst. You must distinguish between minor procedural choices and significant differences in core steps. Assume that most solutions might share a high-level strategy; your goal is to find answers that execute core steps in a meaningfully different way.

**Defining Methodological Difference (Your Core Criteria):**

**What IS NOT a Significant Difference (Methodologically Similar):**

- **Order of Calculation:** Calculating value A then B, versus B then A, before combining them in the same way.
- **Algebraic Equivalence:** Using the form $(a + b)^2$ versus $a^2 + 2ab + b^2$.
- **Variable Naming or Notation:** Using $n$ vs $x$.
- **Choice of Standard Procedural Equivalents:** One summary describes solving a system of equations using **substitution**, while the other uses **elimination**. These are considered standard, interchangeable procedures within the same overall algebraic approach.
- **Rigorous Proof vs. Heuristic Assumption:** If the overall strategy is the same, simply proving a result versus assuming it does not constitute a diverse approach. Both are still following the same high-level logical path.

**What IS a Significant Difference (Methodologically Diverse):**

- This difference represents a **completely distinct, independent, high-level strategic choice** that fundamentally alters the entire problem-solving path from beginning to end.
- **Example 1 (Different Overall Framework):** One solution to a geometry problem uses **coordinate geometry**, another uses **synthetic geometry**, and a third uses **vector analysis**.
- **Example 2 (Completely Different Logical Path):** To solve a counting problem, one answer uses **direct casework**, another uses **complementary counting**, and a third uses a **recurrence relation**.
- **Example 3 (Change in Analytical Tool):** A solution to an optimization problem uses **calculus**, a second uses **inequalities** (like AM-GM), and a third uses **linear programming**.

---

2. CONTENT TO ANALYZE

---

**Problem:**
{question}

**Proposed Solutions (Summarized by Logical Steps):**
{summaries_text}

---

3. OUTPUT REQUIREMENT

---

Based on the final criteria review, classify the diversity of the solutions.
**Output Requirement:**
Immediately after your classification, provide your final answer in a strict JSON format

---

within a special block. The JSON should be a single integer, either `1` or `2`. Do not provide any other text.

Example of Final Output Structure for a **Diverse** problem:

`//boxed{{2}}`

Example of Final Output Structure for a **Not Diverse** problem:

`//boxed{{1}}`

Begin Analysis and Provide Output:

This classification process is applied to the 1,600 high-quality problems in our candidate pool, yielding the diversity distribution statistics reported in Section 3.2.

## C    CASE STUDIES AND ANALYSIS OF THE RPD METRIC

To provide a deeper insight into the effectiveness of our RPD metric, this section presents both a statistical overview and concrete, illustrative examples comparing it against a standard baseline.

### C.1    STATISTICAL DISTRIBUTION OF DIVERSITY SCORES

We first analyze the overall behavior of RPD compared to a common baseline. The baseline method calculates the cosine distance between the embeddings of the full, raw solution texts. We sampled 100 problems from our candidate pool and computed all pairwise diversity scores for their solutions using both methods, resulting in a total of 8,986 data points (i.e., solution pairs) for each distribution.

Figure 3 illustrates the resulting score distributions. The baseline scores are heavily concentrated in a very narrow range near zero (0.00–0.04). This indicates that full-text embeddings are largely insensitive to the underlying reasoning structure, assigning nearly identical low-diversity scores to most pairs and failing to distinguish between subtle and significant strategic differences. In contrast, our RPD metric produces a much wider and more uniform distribution. This indicates that RPD possesses significantly higher resolution and sensitivity, allowing it to capture a continuous spectrum of strategic differences, from the subtle to the substantial.

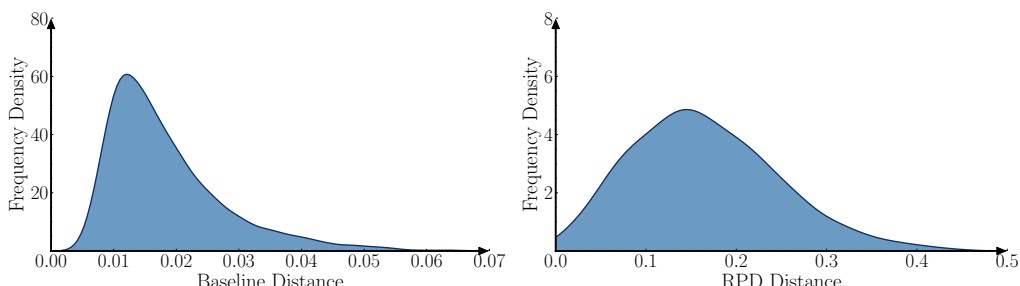

Figure 3: Distribution of pairwise diversity scores on 100 problems for the baseline (left) and our RPD metric (right). RPD provides a significantly better-separated distribution.

## C.2 ILLUSTRATIVE EXAMPLES

The following case studies provide concrete examples of this phenomenon.

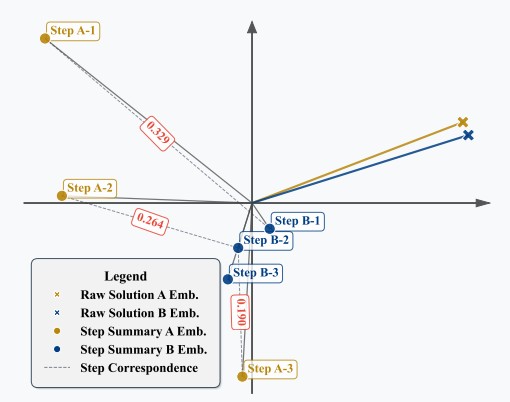
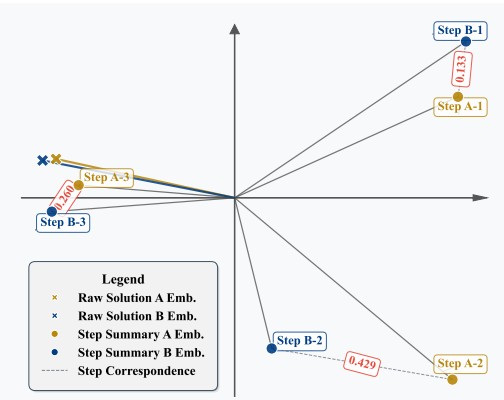

(a) Case Study 1
Raw Emb. Distance: 0.015 (Percentile: 44.46%)
RPD Distance: 0.259 (Percentile: 86.92%)

(b) Case Study 2
Raw Emb. Distance: 0.016 (Percentile: 52.14%)
RPD Distance: 0.274 (Percentile: 90.44%)

Figure 4: PCA visualization of raw solution and step summary embeddings. The step embeddings for the two solutions occupy distinct regions of the space, reflecting a strategic diversity that our RPD metric correctly identifies. In contrast, the raw solution embeddings are nearly collinear, causing the baseline method to fail to distinguish them.

**Case Study 1: Summaries for Figure 4a**

- **Question:** Find the constant term in the polynomial $(x^2 + 2x + 1)(x^2 - 3x - 2) + (x^2 - 2x - 1)(x^2 + 4x + 3)$ after it is factored.

- **Solution A (Full Expansion):**
  - Step 1: Expand each trinomial product using the distributive property.
  - Step 2: Add the expanded polynomials together and combine like terms.
  - Step 3: Extract the constant term from the resulting polynomial.

- **Solution B (Constant Term Shortcut):**
  - Step 1: Determine the constant term of each product by multiplying the constant terms of the individual polynomials.
  - Step 2: Add the constant terms from each product to find the constant term of the entire expression.
  - Step 3: Verify that the constant term remains unchanged when the polynomial is factored.

**Case Study 2: Summaries for Figure 4b**

- **Question:** Determine the largest real value of $a$ such that the equation
  $$ax = x^3 + 1$$
  has a real solution.

- **Solution A (Calculus Approach):**
  - Step 1: Rewrite the equation to express $a$ as a function of $x$, $a = \frac{x^3+1}{x}$.
  - Step 2: Find the critical points of the function $f(x) = \frac{x^3+1}{x}$ by taking its derivative and setting it to zero.
  - Step 3: Evaluate the function at the critical point to find the value of $a$ where the equation has a double root, ensuring the largest $a$ for which the equation has a real solution.

- **Solution B (Geometric Interpretation):**
    - Step 1: Interpret the equation as the intersection of a line $y = ax$ and a curve $y = x^3 + 1$.
    - Step 2: Set the derivative of the cubic function equal to the slope of the line to find the point of tangency.
    - Step 3: Solve the system of equations to find the largest real value of $a$ corresponding to the tangency condition.

**Analysis:** The two case studies in Figure 4 illustrate a consistent pattern where our RPD metric succeeds and the baseline fails. In both examples, the solution pairs employ fundamentally different strategies. The baseline Raw Embedding Distance assigns very low scores (0.015 and 0.016) that correspond to mediocre percentiles (44-52%). This indicates the method is unable to reliably distinguish these solutions from the vast majority of superficially similar pairs. In stark contrast, our RPD metric assigns high scores (0.259 and 0.274) that fall into high percentiles (87-90%), correctly identifying the significant strategic divergence. The PCA visualizations visually corroborate this finding: the well-separated step embeddings in both (a) and (b) confirm that the solutions follow distinct reasoning paths, a fact that only RPD consistently captures.

# D    EXPERIMENT IMPLEMENTATION DETAILS

## D.1    RPD METRIC EVALUATION (DETAILS FOR SEC. 4.1)

In this section, we provide the implementation details for the RPD metric evaluation, including the prompts used for the LLM-based baseline and the evaluation judge.

### D.1.1    PROMPT FOR THE LLM-SELECTION BASELINE

To create the "LLM Selection" baseline, we prompt the Qwen3-14B model to identify the most diverse pair of solutions from all available candidates for a given problem. The prompt is designed to encourage a focus on strategic differences rather than superficial text variations.

---

**Prompt for Selecting the Most Methodologically Diverse Solution Pair**

You are a master mathematician and an expert in pedagogical analysis. Your task is to analyze multiple proposed solutions for a given problem and select a single pair of answers that represents the maximum possible methodological diversity. If no such pair exists, you must indicate this.

Your goal is to identify **one pair of answers** that represents a significant difference in a core step or sub-methodology. If all solutions follow a fundamentally similar strategy, your answer will be to select **"No"**.

---

1. YOUR ANALYSIS FRAMEWORK & CORE CRITERIA

---

Your primary task is to act as a discerning analyst. You must distinguish between minor procedural choices and significant differences in core steps. Assume that most solutions might share a high-level strategy; your goal is to find answers that execute core steps in a meaningfully different way.

**Defining Methodological Difference (Your Core Criteria):**

**What IS NOT a Significant Difference (Methodologically Similar):**

- **Order of Calculation:** Calculating value A then B, versus B then A, before combining them in the same way.
- **Algebraic Equivalence:** Using the form $(a + b)^2$ versus $a^2 + 2ab + b^2$.
- **Variable Naming or Notation:** Using $n$ vs $x$.
- **Choice of Standard Procedural Equivalents:** One summary describes solving a system of equations using **substitution**, while the other uses **elimination**. These are considered standard, interchangeable procedures within the same overall algebraic approach.
- **Rigorous Proof vs. Heuristic Assumption:** If the overall strategy is the same, simply proving a result versus assuming it does not constitute a diverse approach. Both are still following the same high-level logical path.

**What IS a Significant Difference (Methodologically Diverse):**

- This difference represents a **completely distinct, independent, high-level strategic choice** that fundamentally alters the entire problem-solving path from beginning to end.
- **Example 1 (Different Overall Framework):** One solution to a geometry problem uses **coordinate geometry**, another uses **synthetic geometry**, and a third uses **vector analysis**.
- **Example 2 (Completely Different Logical Path):** To solve a counting problem, one answer uses **direct casework**, another uses **complementary counting**, and a third uses a **recurrence relation**.

---

- **Example 3 (Change in Analytical Tool):** A solution to an optimization problem uses **calculus**, a second uses **inequalities** (like AM-GM), and a third uses **linear programming**.

---

2. CONTENT TO ANALYZE

---

**Problem:**
{question}

**Proposed Solutions (Summarized by Logical Steps):**
{summaries_text}

---

3. FINAL INSTRUCTIONS & OUTPUT REQUIREMENT

---

**Your Task:**
Based on the final criteria review, analyze the solutions and make one of two possible determinations:

1. Identify the single pair of answers with the maximum methodological diversity.

2. Conclude that no pair meets the criteria for significant diversity, meaning all solutions follow a fundamentally similar approach.

**Step 1: Brief Comparative Analysis**

- **If you find a diverse pair:** Write a single, brief paragraph. Do not summarize each solution individually. Instead, group the solutions by common methodology and justify your selection of the most diverse pair. For example: "Solution A uses direct casework, while Solution B uses complementary counting. This represents the most significant methodological difference."

- **If you do NOT find a diverse pair:** Write a single, brief paragraph explaining why. State that all solutions follow a similar core strategy and briefly describe that common approach. For example: "All solutions utilize a system of linear equations to solve for the variables. While they use different methods like substitution or elimination, this does not represent a significant strategic divergence. Therefore, no pair is methodologically diverse."

**Step 2: Final JSON Output** Immediately after your brief analysis paragraph, provide your final answer in a strict JSON format within a special block.

- **If a diverse pair is found:** The JSON should be a list containing the single selected answer ID pair.

- **If no diverse pair is found:** The JSON should contain the string "No" within the list structure to maintain format consistency.

**Example of Final Output Structure (Diverse Pair):**
[Your brief analysis justifying the choice...]

```
//boxed_json{{[[id_A, id_B]]}}
```

**Example of Final Output Structure (No Diverse Pair):**
[Your brief analysis explaining the lack of diversity...]

```
//boxed_json{{[["No"]]}}
```

Begin Analysis and Provide Output:

### D.1.2 THE LLM EVALUATION JUDGE

To automate the calculation of the "success rate," a LLM Judge (Qwen3-14B) is used to provide a final verdict on the diversity of a solution pair selected by a given method (e.g., RPD, Raw Emb., etc.). This section details the prompt used to guide the judge and the study conducted to validate its alignment with human judgment.

**Judge Prompt.** The judge is provided with the problem statement and a single pair of solutions. Its task is to assess whether the two solutions employed genuinely different problem-solving strategies. The prompt explicitly instructs the judge to ignore minor differences in wording or calculation and focus on the core reasoning approach.

---

**Prompt for Methodological Similarity Rating**

You are an expert Answer Analysis Assistant, specializing in understanding and comparing the logic and methodology behind problem-solving. Your task is to receive a question, two full answers with their summaries, and rate them strictly based on the similarity of their **methodology**.

**Note:** Based on your prior analysis, you should assume that all proposed solutions for this problem follow a similar high-level strategy. Your task is to find and rate the **methodological diversity within this shared high-level strategy**.

---

RATING CRITERIA

---

Your task is to determine if the two answers are **Methodologically Similar** or **Methodologically Diverse** based on the criteria below, and assign a corresponding rating.

- **Rating 1 (Methodologically Similar):** The two answers are considered similar if the differences are superficial. The following are **NOT** considered significant methodological differences:

  - **Order of Calculation:** Calculating value A then B, versus B then A, before combining them in the same way.
  - **Algebraic Equivalence:** Using the form `(a+b)^2` versus `a^2 + 2ab + b^2`.
  - **Variable Naming or Notation:** Using $n$ vs $x$.
  - **Choice of Standard Procedural Equivalents:** One summary describes solving a system of equations using **substitution**, while the other uses **elimination**. These are considered standard, interchangeable procedures within the same overall algebraic approach.
  - **Rigorous Proof vs. Heuristic Assumption:** If the overall strategy is the same, simply proving a result versus assuming it does not constitute a diverse approach. Both are still following the same high-level logical path.

- **Rating 2 (Methodologically Diverse):** The two answers are considered diverse if the difference represents a **completely distinct, independent, high-level strategic choice** that fundamentally alters the entire problem-solving path from beginning to end.

  - **Example 1 (Different Overall Framework):** One solution to a geometry problem uses **coordinate geometry**, another uses **synthetic geometry**, and a third uses **vector analysis**.
  - **Example 2 (Completely Different Logical Path):** To solve a counting problem, one answer uses **direct casework**, another uses **complementary counting**, and a third uses a **recurrence relation**.
  - **Example 3 (Change in Analytical Tool):** A solution to an optimization problem uses **calculus**, a second uses **inequalities** (like AM-GM), and a third uses **linear programming**.

---

---

OUTPUT REQUIREMENT

---

First, provide a detailed analysis explaining the methodological similarities and differences based on the criteria above. After your analysis is complete, provide the final rating on a new line in the format //boxed{{rating_number}}. **DO NOT ONLY GIVE OUT YOUR RATE!**

---

**Begin Analysis:**

**[Question]:**
{question}

**[Answer A]:**
{answer_a}

**[Answer A summary]:**
{summary_a}

**[Answer B]:**
{answer_b}

**[Answer B summary]:**
{summary_b}

---

**Validation.** To ensure the reliability of the LLM Judge used as our primary evaluation criterion in Sec. 4.1, we conduct an alignment study with human annotations.

To validate the judge, we first construct a dedicated test set. Human annotators select 100 pairs of solutions from our candidate pool, creating a balanced ground-truth dataset composed of 50 pairs with semantically *diverse* reasoning paths and 50 pairs with the *same* underlying reasoning path.

The LLM Judge is then tasked with making a binary diversity judgment on each of these 100 pairs. The results are presented in the confusion matrix in Table 7. Overall, the LLM Judge achieves an accuracy of 78%, demonstrating a strong alignment with human judgment and performing significantly better than a random baseline (50%). We observe that the judge is quite effective at identifying truly diverse pairs (Recall 82%), though it is slightly prone to false positives (classifying similar paths as diverse). This level of agreement validates our use of the LLM Judge as a reliable automated proxy for evaluating reasoning diversity in our main experiment.

Table 7: Confusion matrix of LLM Judge verdicts against human annotations on 100 solution pairs.

|  |  | LLM Judge Verdict | |
|---|---|---|---|
|  |  | Diverse | Same |
| **Human** | **Diverse** | 41 (TP) | 9 (FN) |
| **Label** | **Same** | 13 (FP) | 37 (TN) |

### D.2 DETAILS FOR MULTI-SOLUTION FINE-TUNING (SEC. 4.2)

This section provides detailed implementation procedures for the main fine-tuning experiment, focusing on how the baseline training sets were constructed. Each method aims to select 100 problems and 3 solutions per problem, but they differ in their core selection strategy.

#### D.2.1 RANDOM SELECTION BASELINE

The **Random 1P3S** baseline was constructed through a naive sampling process. We first randomly selected 100 problems from our 1,600-problem candidate pool without replacement. For each of these 100 problems, we then randomly selected 3 of its available solutions to form the training data. This method serves as a fundamental baseline to measure the benefits of any systematic diversity-driven selection.

### D.2.2 LLM SELECTION BASELINE

This baseline leverages the powerful Qwen3-14B model to simulate an expert's judgment in a two-stage curation process. First, the LLM performs a binary classification to identify whether a problem's solutions are methodologically diverse. We then selected 100 problems that were positively classified as containing diverse solution methods. Second, for these selected problems, the LLM is prompted again to choose the set of 3 solutions that are maximally distinct from each other. The specific prompts for each stage are provided below.

---

**Prompt for Problem Diversity Classification**

You are a master mathematician and an expert in pedagogical analysis. Your task is to classify a problem based on the methodological diversity of its proposed solutions.
Your goal is to perform a binary classification:

- **Class 2 (Diverse):** If the provided solution summaries showcase more than one distinct core methodology.

- **Class 1 (Not Diverse):** If all solutions use the same core methodology, or if the differences are only superficial (e.g., a different order of calculation, or using standard procedural equivalents like substitution vs. elimination).

---

1. YOUR ANALYSIS FRAMEWORK & CORE CRITERIA

Your primary task is to act as a discerning analyst. You must distinguish between minor procedural choices and significant differences in core steps. Assume that most solutions might share a high-level strategy; your goal is to find answers that execute core steps in a meaningfully different way.

**Defining Methodological Difference (Your Core Criteria):**

**What IS NOT a Significant Difference (Methodologically Similar):**

- **Order of Calculation:** Calculating value A then B, versus B then A, before combining them in the same way.

- **Algebraic Equivalence:** Using the form $(a + b)^2$ versus $a^2 + 2ab + b^2$.

- **Variable Naming or Notation:** Using $n$ vs $x$.

- **Choice of Standard Procedural Equivalents:** One summary describes solving a system of equations using **substitution**, while the other uses **elimination**. These are considered standard, interchangeable procedures within the same overall algebraic approach.

- **Rigorous Proof vs. Heuristic Assumption:** If the overall strategy is the same, simply proving a result versus assuming it does not constitute a diverse approach. Both are still following the same high-level logical path.

**What IS a Significant Difference (Methodologically Diverse):**

- This difference represents a **completely distinct, independent, high-level strategic choice** that fundamentally alters the entire problem-solving path from beginning to end.

- **Example 1 (Different Overall Framework):** One solution to a geometry problem uses **coordinate geometry**, another uses **synthetic geometry**, and a third uses **vector analysis**.

- **Example 2 (Completely Different Logical Path):** To solve a counting problem, one answer uses **direct casework**, another uses **complementary counting**, and a third uses a **recurrence relation**.

- **Example 3 (Change in Analytical Tool):** A solution to an optimization problem uses **calculus**, a second uses **inequalities** (like AM-GM), and a third uses **linear programming**.

---

2. CONTENT TO ANALYZE

**Problem:**
{question}

**Proposed Solutions (Summarized by Logical Steps):**
{summaries_text}

---

3. OUTPUT REQUIREMENT

Based on the final criteria review, classify the diversity of the solutions.

**Output Requirement:** Immediately after your classification, provide your final answer in a strict JSON format within a special block. The JSON should be a single integer, either 1 or 2. Do not provide any other text.

**Example of Final Output Structure for a Diverse problem:**
`//boxed{{2}}`

**Example of Final Output Structure for a Not Diverse problem:**
`//boxed{{1}}`

**Begin Analysis and Provide Output:**

---

**Prompt for Diverse Solution Selection**

You are a master mathematician and an expert in pedagogical analysis. Your task is to analyze multiple proposed solutions for a given problem and select a set of {num_to_select} answers that, as a set, represents the maximum possible methodological diversity.
Your goal is to identify a single set of {num_to_select} answers where each chosen answer has a significant methodological difference from every other answer in the set. Think of it as finding a set of three solutions that are all mutually distinct in their core approach.

---

1. YOUR ANALYSIS FRAMEWORK & CORE CRITERIA

Your primary task is to act as a discerning analyst. You must distinguish between minor procedural choices and significant differences in core steps. Assume that most solutions might share a high-level strategy; your goal is to find answers that execute core steps in a meaningfully different way.

**Defining Methodological Difference (Your Core Criteria):**

**What IS NOT a Significant Difference (Methodologically Similar):**

- **Order of Calculation:** Calculating value A then B, versus B then A, before combining them in the same way.
- **Algebraic Equivalence:** Using the form $(a + b)^2$ versus $a^2 + 2ab + b^2$.
- **Variable Naming or Notation:** Using $n$ vs $x$.
- **Choice of Standard Procedural Equivalents:** One summary describes solving a system of equations using **substitution**, while the other uses **elimination**. These are considered standard, interchangeable procedures within the same overall algebraic approach.
- **Rigorous Proof vs. Heuristic Assumption:** If the overall strategy is the same, simply proving a result versus assuming it does not constitute a diverse approach. Both are still following the same high-level logical path.

**What IS a Significant Difference (Methodologically Diverse):**

- This difference represents a **completely distinct, independent, high-level strategic choice** that fundamentally alters the entire problem-solving path from beginning to end.

- **Example 1 (Different Overall Framework):** One solution to a geometry problem uses **coordinate geometry**, another uses **synthetic geometry**, and a third uses **vector analysis**.
- **Example 2 (Completely Different Logical Path):** To solve a counting problem, one answer uses **direct casework**, another uses **complementary counting**, and a third uses a **recurrence relation**.
- **Example 3 (Change in Analytical Tool):** A solution to an optimization problem uses **calculus**, a second uses **inequalities** (like AM-GM), and a third uses **linear programming**.

---

2. CONTENT TO ANALYZE

**Problem:**
{question}

**Proposed Solutions (Summarized by Logical Steps):**
{summaries_text}

---

3. FINAL INSTRUCTIONS & OUTPUT REQUIREMENT

Your Task: Based on the final criteria review, analyze the solutions.

**Step 1: Brief Comparative Analysis** First, write a single, brief paragraph for your analysis. Do not summarize each solution individually. Instead, group the solutions by common methodology and justify your selection of the set of {num_to_select} most diverse answers. For example: Solutions A and C use direct casework, while Solution B uses complementary counting, and Solution D uses a geometric approach. The most diverse set is [A, B, D] as it captures these three distinct methods.

**Step 2: Final JSON Output** Immediately after your brief analysis paragraph, provide your final answer in a strict JSON format within a special block. The JSON should be a list containing the {num_to_select} selected answer IDs.

**Example of Final Output Structure:**
[Your brief analysis...]
//boxed_json{{[id_A, id_B, id_C]}}

**Begin Analysis and Provide Output:**

### D.2.3 EMBEDDING-BASED BASELINE

To rigorously evaluate the effectiveness of our RPD metric, we compare it against two baseline distance metrics. For a fair comparison, all training datasets—both for our method and the baselines—are constructed using the identical **two-stage data curation framework** detailed previously. This framework consists of **Stage 1: Problem Selection** (Algorithm 2) and **Stage 2: Greedy Solution Selection** (Algorithm 3).

The sole difference between our method and the baselines is the specific pairwise distance function, $\mathcal{D}(S_i, S_j)$, that is plugged into this framework. The baseline metrics are defined below.

**Raw Solution Cosine Distance** ($D_{\mathbf{raw}}$) This baseline metric computes the cosine distance between the embedding vectors of the complete solution texts. For all embedding tasks, we use the Qwen3-Embedding-8B model. Let $\mathcal{M}_{\text{embed}}$ be this model.

$$D_{\text{raw}}(S_i, S_j) = 1 - \frac{\mathcal{M}_{\text{embed}}(S_i) \cdot \mathcal{M}_{\text{embed}}(S_j)}{\|\mathcal{M}_{\text{embed}}(S_i)\|\|\mathcal{M}_{\text{embed}}(S_j)\|}$$

**Summary Cosine Distance** ($D_{\mathbf{summary}}$) This baseline first concatenates the step-level summaries for a solution to form a single composite summary text. The diversity is then computed as the cosine

distance between the embeddings of these composite summaries.

$$D_{\text{summary}}(S_i, S_j) = 1 - \frac{\mathcal{M}_{\text{embed}}(\text{Summary}_{\text{comp}}(S_i)) \cdot \mathcal{M}_{\text{embed}}(\text{Summary}_{\text{comp}}(S_j))}{\|\mathcal{M}_{\text{embed}}(\text{Summary}_{\text{comp}}(S_i))\| \|\mathcal{M}_{\text{embed}}(\text{Summary}_{\text{comp}}(S_j))\|}$$

Based on the framework detailed previously, we generate three distinct training datasets:

- **Ours (RPD)**: Constructed by applying the two-stage framework with our proposed RPD metric ($D_{\text{RPD}}$).
- **Raw Emb.**: Constructed using the same framework but with the $D_{\text{raw}}$ metric.
- **Summary Emb.**: Constructed using the same framework but with the $D_{\text{summary}}$ metric.

# E  EXPERIMENT RESULTS

This appendix presents the complete experimental results for both models. The tables are structured to clearly distinguish between the pre-trained baseline model, fine-tuning with a one-problem-one-solution (1P1S) paradigm, and fine-tuning with a one-problem-three-solution (1P3S) paradigm.

## E.1  COMPLETE RESULTS FOR QWEN3-4B-BASE MODEL

The following tables present the comprehensive performance of the Qwen3-4B-Base model on the AIME24 and Olympiad Benchmarks, which complements the MATH500 Level 5 results from the main paper. As shown in Table 8, our **RPD** method demonstrates a significant performance improvement by adopting the *one problem, multiple solutions* paradigm. It elevates the `pass@16` score to **35.83%** on AIME24, surpassing the standard 1P1S baseline (Random 1P1S) by an impressive **4.99** percentage points. Furthermore, our RPD-guided curation strategy also proves its superiority over other 1P3S methods, with its `pass@16` score outperforming the next-best baseline (Random 1P3S) by **2.50** percentage points on the same benchmark. This pattern holds for the Olympiad Bench (Table 9), where our method achieves a leading `pass@16` score of **68.11%**, which is **1.56** percentage points higher than the 1P1S baseline and **0.75** percentage points higher than the best alternative 1P3S method. These results provide strong evidence for the effectiveness of our approach in both paradigm and data curation strategy.

Table 8: Full comparison on the **AIME24** benchmark using the **Qwen3-4B-Base** model.

| Paradigm | Method | pass@1 (%) | pass@2 (%) | pass@4 (%) | pass@8 (%) | pass@16 (%) |
|---|---|---|---|---|---|---|
| Pre-trained | Base | 8.34 | 13.33 | 16.67 | 21.67 | 27.50 |
| 1P1S | Random 1P1S | **14.17** | 18.33 | 23.33 | 26.67 | 30.84 |
| 1P3S | Random 1P3S | 9.17 | 12.50 | 19.17 | 28.34 | 33.33 |
| | Raw Emb. | 12.50 | 16.67 | 20.00 | 25.84 | 33.33 |
| | Summary Emb. | 10.00 | 12.50 | 17.50 | 25.00 | 29.17 |
| | LLM Selection | 10.83 | 15.84 | 20.83 | 25.83 | 30.83 |
| | **Ours (RPD)** | **14.17** | **19.17** | **25.83** | **30.00** | **35.83** |

Table 9: Full comparison on the **Olympiad Bench** using the **Qwen3-4B-Base** model.

| Paradigm | Method | pass@1 (%) | pass@2 (%) | pass@4 (%) | pass@8 (%) | pass@16 (%) |
|---|---|---|---|---|---|---|
| Pre-trained | Base | 39.54 | 47.11 | 53.56 | 61.13 | 65.95 |
| 1P1S | Random 1P1S | **42.43** | 49.18 | 55.49 | 61.43 | 66.55 |
| 1P3S | Random 1P3S | 40.13 | 50.15 | 56.75 | 62.61 | 67.36 |
| | Raw Emb. | 39.91 | 47.48 | 56.38 | 61.42 | 66.62 |
| | Summary Emb. | 40.88 | 49.78 | 57.05 | 62.69 | 66.92 |
| | LLM Selection | 39.62 | 48.30 | 56.60 | 62.83 | 67.06 |
| | **Ours (RPD)** | 41.92 | **51.19** | **57.50** | **63.06** | **68.11** |

## E.2  COMPLETE RESULTS FOR QWEN2.5-3B MODEL

To demonstrate the robustness and generalizability of our findings, we also fine-tuned the Qwen2.5-3B model. Specifically, we employed supervised fine-tuning using 4-bit QLoRA (rank=16, alpha=32), training the model for **15 epochs** in BF16 precision. We utilized the AdamW optimizer with a cosine learning rate scheduler, setting the peak learning rate to $4 \times 10^{-5}$. We then evaluated its performance across the same three benchmarks (Tables 10, 11, and 12).

The results consistently reaffirm our core hypothesis. For instance, on the AIME24 benchmark (Table 10), our **RPD** method's advantage is particularly pronounced when evaluating with a larger sample set. Focusing on the key `pass@16` metric, our approach achieves a score of **22.50%**. This represents a substantial 5.00 percentage point improvement over the 1P1S baseline and demonstrates a clear advantage over other multi-solution strategies, outperforming the next-best 1P3S methods

by 0.83 percentage points. The outperformance on AIME24 exemplifies a consistent trend also observed on the MATH500 and Olympiad benchmarks, which solidifies the conclusion that our RPD-guided data curation is a general and effective technique for enhancing Test-Time Scaling.

Table 10: Full comparison on the **AIME24** benchmark using the **Qwen2.5-3B** model.

| Paradigm | Method | pass@1 (%) | pass@2 (%) | pass@4 (%) | pass@8 (%) | pass@16 (%) |
|---|---|---|---|---|---|---|
| Pre-trained | Base | 4.17 | 4.17 | 10.00 | 16.67 | 16.67 |
| 1P1S | Random 1P1S | 4.17 | 8.34 | 10.00 | 13.33 | 17.50 |
| 1P3S | Random 1P3S | 6.67 | 8.33 | 14.17 | 18.33 | 20.00 |
| | Raw Emb. | 5.84 | 8.34 | 14.17 | 18.33 | 20.83 |
| | Summary Emb. | 3.33 | 6.67 | 13.33 | 18.33 | 21.67 |
| | LLM Selection | 2.50 | 5.00 | 13.33 | 16.67 | 21.67 |
| | **Ours (RPD)** | **7.50** | **10.00** | **15.00** | **20.00** | **22.50** |

Table 11: Full comparison on the **MATH500 Level 5** benchmark using the **Qwen2.5-3B** model.

| Paradigm | Method | pass@1 (%) | pass@2 (%) | pass@4 (%) | pass@8 (%) | pass@16 (%) |
|---|---|---|---|---|---|---|
| Pre-trained | Base | 23.70 | 32.65 | 43.84 | 55.60 | 63.62 |
| 1P1S | Random 1P1S | 29.11 | 41.05 | 51.31 | 60.45 | 67.73 |
| 1P3S | Random 1P3S | **31.72** | **42.91** | 50.94 | 60.82 | 68.28 |
| | Raw Emb. | 28.92 | 40.86 | 51.31 | 60.08 | 69.22 |
| | Summary Emb. | 27.05 | 38.06 | 51.12 | 60.26 | 67.35 |
| | LLM Selection | 27.61 | 37.87 | 49.82 | 60.26 | 67.91 |
| | **Ours (RPD)** | 28.55 | 40.30 | **51.49** | **61.20** | **69.97** |

Table 12: Full comparison on the **Olympiad Bench** using the **Qwen2.5-3B** model.

| Paradigm | Method | pass@1 (%) | pass@2 (%) | pass@4 (%) | pass@8 (%) | pass@16 (%) |
|---|---|---|---|---|---|---|
| Pre-trained | Base | 21.81 | 30.27 | 37.54 | 45.55 | 51.93 |
| 1P1S | Random 1P1S | 19.14 | 27.45 | 35.68 | 45.48 | 52.89 |
| 1P3S | Random 1P3S | 22.33 | 30.79 | 39.10 | 47.11 | 53.93 |
| | Raw Emb. | 22.03 | 30.05 | 39.10 | 46.52 | 52.90 |
| | Summary Emb. | **22.85** | **31.34** | 38.95 | 46.63 | 53.82 |
| | LLM Selection | 21.96 | 30.79 | **39.25** | 47.11 | 53.94 |
| | **Ours (RPD)** | 20.40 | 30.19 | 39.10 | **47.18** | **54.16** |

### E.3 Ablation Study: Performance at a Larger Scale (1500 Samples)

To assess the scalability of our 1PNS paradigm, we conducted an additional experiment by increasing the total training data size to 1,500 samples. This study compares our diversity-driven (500Q, 3A) configuration against a traditional (1500Q, 1A) baseline. The results, presented in Table 13, show that our approach maintains a significant advantage, particularly on the pass@k metrics. This confirms that the benefits of multi-solution fine-tuning are robust and effective even at a larger data scale.

Table 13: Performance comparison of our 1PNS approach against the 1P1S baseline across three mathematical reasoning benchmarks.

| Benchmark | Method | pass@1 (%) | pass@2 | pass@4 | pass@8 | pass@16 |
|---|---|---|---|---|---|---|
| AIME24 | Random (1P1S) | **13.33** | 16.67 | 18.34 | 25.00 | 30.00 |
| | RPD (1P3S) | 12.50 | **19.17** | **22.50** | **25.84** | **35.00** |
| MATH500 Level 5 | Random (1P1S) | **52.80** | **62.32** | **68.66** | 72.58 | 75.94 |
| | RPD (1P3S) | 51.49 | 58.96 | 66.61 | **72.95** | **78.18** |
| Olympiad Bench | Random (1P1S) | 39.77 | **49.48** | 55.86 | 62.24 | 66.62 |
| | RPD (1P3S) | **39.99** | 49.33 | **57.20** | **63.21** | **67.51** |

