# OpenReview forum: "Reasoning Path Divergence: A New Metric and Curation Strategy to Unlock LLM Diverse Thinking"
_ICLR.cc/2026/Conference — ICLR 2026 Conference Withdrawn Submission_

### Official Review · Reviewer_xNim · 2025-10-20

**Soundness:** 2
**Presentation:** 3
**Contribution:** 2
**Rating:** 4
**Confidence:** 4

**Summary:**

To address the low diversity issue of traditional one-problem-one-solution fine-tuning (1P1S) paradigm, this paper proposes an effective one-problem-multiple-solutions fine-tuning method based on a newly proposed metric-Reasoning Path Divergence (RPD). Specifically, after obtaining multiple solutions to a problem, the authors first leverage a LLM to decompose each solution into several summarized core steps. Then, for each two solutions, the authors calculate the semantic distance between each step in one solution to all steps in another solution. The RPD score is calculated as the average calculated semantic distances over all steps in that solution. Selecting solutions with higher RPD scores with other solutions can bring more diversity to the dataset, and fine-tuning the model on this dataset helps to improve the Pass@K metric during inference.

**Strengths:**

(1) The paper is generally well-written, and the structure is clear.

(2) The proposed metric, Reasoning Path Divergence, although somewhat complex and intricate, remains quite reasonable and intuitive.

(3) The authors try to conduct comprehensive ablations to study the effectiveness of the method.

**Weaknesses:**

1. First, the calculation process of RPD is quite complex and unstable. Its computation heavily relies on decomposing the original solutions using an external LLM, a process that is not always accurate and reproducible, and is highly dependent on the LLM's instruction-following capabilities and designed prompts.

2. The experiments are conducted on a very small sample size (300 samples), which makes the results and findings potentially unreliable. The authors should perform experiments on a larger-scale dataset.

3. The experiments are only conducted on Qwen models, lacking the validation of the method's generalizability on other model architectures, such as Llama.

4. The number of evaluation sets is limited. The sample sizes of AIME24 and MATH500-Level are limited. The authors are encouraged to perform thorough evaluations on more test sets.

5. The included baselines are naive and weak. A strong baseline should be training the model on all the original, unfiltered data. If similar or better response diversity and performance can be achieved without filtering, then the need for extra data filtering seems less justified.

6. I am quite interested in whether the proposed method or trained model in this paper can bring improvements in subsequent RL training, leveraging the higher Pass@K results.

**Questions:**

1. The authors compared the performance of various methods after implementing different data filtering measures. However, a direct baseline would involve using the original, unfiltered data for SFT without any additional data screening. If similar or better response diversity and performance can be achieved without filtering, then the need for extra data filtering seems less justified.

2. Why choose LoRA fine-tuning instead of full parameter fine-tuning?

---

> ### Author Response · Authors · 2025-11-20
>
> Thank you for your time and effort in reviewing our work. Below we respond to the comments in **Weaknesses (W)** and **Questions (Q)**.
>
> ---
>
> **W1: Stability of the RPD metric calculation.**
>
> Thank you for your very thoughtful consideration regarding the stability of our metric calculation. To test the robustness of our RPD pipeline, we conducted a new experiment using a significantly *weaker* model, **Qwen2.5-7B-Instruct**, for the "Reasoning Step Extraction" phase (Sec 3.1), while keeping the main model (Qwen3-4B-Base) and the training process the same.
>
> As shown in Table 1, the performance of our method using the 7B summarizer is comparable to using much stronger 14B summarizer across different k-values, and crucially, both outperform the Random 1P1S baseline at larger values of k in pass@k. This result demonstrates that our RPD pipeline is not dependent on a specific large-scale model. The detailed, structured prompt (Appendix A.1) enables even smaller, instruction-tuned models to perform the summarization task reliably, confirming the stability of our method.
>
> **Table 1: RPD Robustness to Summarizer Model**
>
> | Benchmark  | Method                             | pass@1 (%) | pass@2 (%) | pass@4 (%) | pass@8 (%) | pass@16 (%) |
> | :--------- | :--------------------------------- | :--------: | :--------: | :--------: | :--------: | :---------: |
> | AIME24     | Random (1P1S)                      | **14.17**  |   18.33    |   23.33    |   26.67    |    30.84    |
> |            | Ours (Qwen3-14B Summary)           | **14.17**  |   19.17    |   25.83    |   **30.00**    |    35.83    |
> |            | Ours (Qwen2.5-7B-Instruct Summary) |   13.33    | **22.50**  | **27.50**  | 29.17  |  **36.67**  |
> | AIME25     | Random (1P1S)                      |   11.67    |   15.83    |   18.33    |   27.50    |    32.50    |
> |            | Ours (Qwen3-14B Summary)           |   10.00    | **18.33**  | **24.17**  | **29.17**  |  **35.83**  |
> |            | Ours (Qwen2.5-7B-Instruct Summary) | **14.17**  | **18.33**  |   23.33    |   27.50    |    33.33    |
> | MATH500 L5 | Random (1P1S)                      |   49.26    |   60.64    |   66.98    |   72.20    |    77.43    |
> |            | Ours (Qwen3-14B Summary)           |   52.61    |   61.57    | **71.64**  |   **75.94**    |    79.29    |
> |            | Ours (Qwen2.5-7B-Instruct Summary) | **53.73**  | **63.81**  |   68.66    | 75.75  |  **79.67**  |
> | Olympiad   | Random (1P1S)                      | **42.43**  |   49.18    |   55.49    |   61.43    |    66.55    |
> |            | Ours (Qwen3-14B Summary)           |   41.92    | **51.19**  | **57.50**  | **63.06**  |  **68.11**  |
> |            | Ours (Qwen2.5-7B-Instruct Summary) |   38.58    |   48.07    |   55.34    |   62.61    |    67.80    |
> | AMC23      | Random (1P1S)                      | **47.50**  |   54.38    |   63.75    |   73.75    |    81.25    |
> |            | Ours (Qwen3-14B Summary)           | **47.50**  | **60.63**  | **71.25**  | **75.63**  |  **81.88**  |
> |            | Ours (Qwen2.5-7B-Instruct Summary) |   46.88    |   56.25    |   65.63    | **75.63**  |    81.25    |
>
> ---
>
> **W2: Scalability on larger datasets.**
>
> Thank you for your suggestion for scalability. We have extended our training sets to 3,000 samples (1,000 problems, 3 solutions each) and compared it to a 3,000-sample 1P1S baseline (due to computational resource constraints,  the current 3,000-sample model was trained for only 1 epoch; all other training settings remained unchanged).
>
> The results on Qwen3-4B-Base (Table 2) confirm that our RPD-curated 1P3S approach continues to outperform the 1P1S baseline at this larger scale.
>
> **Table 2: Performance comparison on larger dataset (3,000 samples).**
>
> | Benchmark  | Method                | pass@1 (%) | pass@2 (%) | pass@4 (%) | pass@8 (%) | pass@16 (%) |
> | :--------- | :-------------------- | :--------- | :--------- | :--------- | :--------- | :---------- |
> | AIME24     | Random (1P1S, 3000)   | **10.00**  | **16.67**  | 20.83      | 25.83      | 32.50       |
> |            | Ours (RPD 1P3S, 3000) | **10.00**  | **16.67**  | **25.00**  | **29.17**  | **35.83**   |
> | AIME25     | Random (1P1S, 3000)   | 9.17       | 14.17      | 19.17      | 23.34      | 32.50       |
> |            | Ours (RPD 1P3S, 3000) | **10.00**  | **15.83**  | **24.17**  | **30.84**  | **36.67**   |
> | MATH500 L5 | Random (1P1S, 3000)   | 50.56      | 58.96      | 66.23      | 73.13      | 77.61       |
> |            | Ours (RPD 1P3S, 3000) | **52.05**  | **60.64**  | **70.34**  | **76.49**  | **80.04**   |
> | Olympiad   | Random (1P1S, 3000)   | 40.06      | 47.77      | 55.64      | 61.57      | 66.91       |
> |            | Ours (RPD 1P3S, 3000) | **40.50**  | **49.26**  | **58.01**  | **63.06**  | **68.84**   |
> | AMC23      | Random (1P1S, 3000)   | 41.25      | **51.25**  | 65.00      | 76.25      | 83.75       |
> |            | Ours (RPD 1P3S, 3000) | **42.50**  | 50.00      | **68.75**  | **78.75**  | **87.50**   |

---

> > ### Author Response · Authors · 2025-11-20
> >
> > **W3: Generalizability to Llama.**
> >
> > Thank you for your valuable suggestion regarding the generalizability of our method. To validate the generalizability of our RPD curation strategy, we conducted new experiments fine-tuning **Llama 3.1 8B Instruct** using 300 samples, with training settings identical to those in Section 4.2.
> >
> > As shown in Table 3, the RPD-curated 1P3S data yields substantial gains over the 1P1S baseline, demonstrating that our method is effective on different model architectures.
> >
> > **Table 3: Generalizability to Llama 3.1 8B Instruct**
> >
> > | Benchmark      | Method          | pass@1 (%) | pass@2 (%) | pass@4 (%) | pass@8 (%) | pass@16 (%) |
> > | :------------- | :-------------- | :--------: | :--------: | :--------: | :--------: | :---------: |
> > | **AIME24**     | Random (1P1S)   |    3.75    |    4.58    |    8.75    |   12.50    |    16.67    |
> > |                | Ours (RPD 1P3S) |  **6.67**  |  **8.75**  | **11.25**  | **15.00**  |  **18.75**  |
> > | **AIME25**     | Random (1P1S)   |    0.00    |    1.67    |    2.50    |    5.83    |    10.00    |
> > |                | Ours (RPD 1P3S) |  **3.33**  |  **4.17**  |  **5.83**  | **10.00**  |  **12.50**  |
> > | **MATH500 L5** | Random (1P1S)   |   17.35    |   26.31    |   35.45    |   44.41    |    52.62    |
> > |                | Ours (RPD 1P3S) | **20.52**  | **28.36**  | **37.69**  | **47.76**  |  **55.23**  |
> > | **Olympiad**   | Random (1P1S)   | **14.99**  |   21.29    |   26.86    |   33.46    |    39.62    |
> > |                | Ours (RPD 1P3S) |   14.54    | **21.59**  | **27.45**  | **33.83**  |  **40.58**  |
> > | **AMC23**      | Random (1P1S)   |   16.25    |   26.88    |   36.25    |   51.88    |    61.88    |
> > |                | Ours (RPD 1P3S) | **23.13**  | **31.25**  | **40.63**  | **53.13**  |  **65.63**  |
> >
> > ---
> >
> > **W4: Evaluation on additional test sets.**
> >
> > Thank you for this valid point. To ensure our evaluation is comprehensive, we have expanded our testbed to include two more mathematical competition level benchmarks: **AIME25** and **AMC23**.
> >
> > Our RPD-curated method shows consistent improvements on these new benchmarks. As shown in Tables 4 and 5, our method outperforms the 1P1S baseline on both Qwen3-4B-Base and Qwen2.5-3B models, particularly at larger values of k in pass@k.
> >
> > **Table 4: Qwen3-4B-Base on New Benchmarks**
> >
> > | Benchmark | Method          | pass@1 (%) | pass@2 (%) | pass@4 (%) | pass@8 (%) | pass@16 (%) |
> > | :-------- | :-------------- | :--------: | :--------: | :--------: | :--------: | :---------: |
> > | AIME25    | Random (1P1S)   | **11.67**  |   15.83    |   18.33    |   27.50    |    32.50    |
> > |           | Ours (RPD 1P3S) |   10.00    | **18.33**  | **24.17**  | **29.17**  |  **35.83**  |
> > | AMC23     | Random (1P1S)   | **47.50**  |   54.38    |   63.75    |   73.75    |    81.25    |
> > |           | Ours (RPD 1P3S) | **47.50**  | **60.63**  | **71.25**  | **75.63**  |  **81.88**  |
> >
> > **Table 5: Qwen2.5-3B on New Benchmarks**
> >
> > | Benchmark | Method          | pass@1 (%) | pass@2 (%) | pass@4 (%) | pass@8 (%) | pass@16 (%) |
> > | :-------- | :-------------- | :--------: | :--------: | :--------: | :--------: | :---------: |
> > | AIME25    | Random (1P1S)   |    1.67    |    4.17    |    8.33    |   12.50    |    14.17    |
> > |           | Ours (RPD 1P3S) |  **3.33**  |  **5.83**  | **12.50**  | **15.00**  |  **18.33**  |
> > | AMC23     | Random (1P1S)   |   30.63    |   41.25    |   51.25    |   62.50    |    75.63    |
> > |           | Ours (RPD 1P3S) | **34.38**  | **46.25**  | **56.88**  | **68.13**  |  **80.00**  |

---

> > > ### Author Response · Authors · 2025-11-20
> > >
> > > **W5 & Q1: Comparison with the strong baseline (unfiltered data).**
> > >
> > > Thanks for suggesting this crucial comparison to justify our data filtering method. We have implemented this "Strong Baseline" by sampling 225 problems from the original dataset and utilizing all 16 available solutions for each, resulting in a total of 3600 training samples. We trained the model on this dataset for 1 epoch to ensure the total number of gradient update steps remains identical to our main experiments (which involved 300 samples trained for 12 epochs), allowing for a fair comparison.
> > >
> > > * **Superior performance.** The results in Table 6 (Qwen3-4B-Base) and Table 7 (Qwen2.5-3B) show that while this Strong Baseline outperforms the standard Random 1P1S baseline in some benchmarks, our RPD-curated (1P3S) method still achieves superior performance, particularly at higher k-values.
> > >
> > > **Table 6: Comparison vs. Strong Baseline (Qwen3-4B-Base)**
> > >
> > > | Benchmark  | Method                             | pass@1 (%) | pass@2 (%) | pass@4 (%) | pass@8 (%) | pass@16 (%) |
> > > | :--------- | :--------------------------------- | :--------: | :--------: | :--------: | :--------: | :---------: |
> > > | AIME24     | Random (1P1S)                      | **14.17**  |   18.33    |   23.33    |   26.67    |    30.84    |
> > > |            | Strong Baseline (Unfiltered  Data) |    6.67    |   12.50    |   20.83    |   25.83    |    29.17    |
> > > |            | Ours (RPD 1P3S)                    | **14.17**  | **19.17**  | **25.83**  | **30.00**  |  **35.83**  |
> > > | AIME25     | Random (1P1S)                      | **11.67**  |   15.83    |   18.33    |   27.50    |    32.50    |
> > > |            | Strong Baseline (Unfiltered  Data) |    8.34    |   15.00    |   19.17    |   22.50    |    33.33    |
> > > |            | Ours (RPD 1P3S)                    |    10.0    | **18.33**  | **24.17**  | **29.17**  |  **35.83**  |
> > > | MATH500 L5 | Random (1P1S)                      |   49.26    |   60.64    |   66.98    |   72.20    |    77.43    |
> > > |            | Strong Baseline (Unfiltered  Data) |   47.20    |   57.46    |   66.60    |   73.51    |    77.80    |
> > > |            | Ours (RPD 1P3S)                    | **52.61**  | **61.57**  | **71.64**  | **75.94**  |  **79.29**  |
> > > | Olympiad   | Random (1P1S)                      | **42.43**  |   49.18    |   55.49    |   61.43    |    66.55    |
> > > |            | Strong Baseline (Unfiltered  Data) |   39.47    |   49.11    |   56.61    |   62.99    |    68.03    |
> > > |            | Ours (RPD 1P3S)                    |   41.92    | **51.19**  | **57.50**  | **63.06**  |  **68.11**  |
> > > | AMC23      | Random (1P1S)                      | **47.50**  |   54.38    |   63.75    |   73.75    |    81.25    |
> > > |            | Strong Baseline (Unfiltered  Data) |   46.25    |   58.75    |   66.88    |   75.00    |    81.25    |
> > > |            | Ours (RPD 1P3S)                    | **47.50**  | **60.63**  | **71.25**  | **75.63**  |  **81.88**  |
> > >
> > > **Table 7: Comparison vs. Strong Baseline (Qwen2.5-3B)**
> > >
> > > | Benchmark  | Method                             | pass@1 (%) | pass@2 (%) | pass@4 (%) | pass@8 (%) | pass@16 (%) |
> > > | :--------- | :--------------------------------- | :--------: | :--------: | :--------: | :--------: | :---------: |
> > > | AIME24     | Random (1P1S)                      |    4.17    |    8.34    |    10.0    |   13.33    |    17.50    |
> > > |            | Strong Baseline (Unfiltered  Data) |    3.34    |    6.67    |   11.67    |   13.33    |    21.67    |
> > > |            | Ours (RPD 1P3S)                    |  **7.50**  | **10.00**  | **15.00**  | **20.00**  |  **22.50**  |
> > > | AIME25     | Random (1P1S)                      |    1.67    |    4.17    |    8.33    |   12.50    |    14.17    |
> > > |            | Strong Baseline (Unfiltered  Data) |  **5.00**  |  **8.34**  |   10.83    |   14.17    |    17.50    |
> > > |            | Ours (RPD 1P3S)                    |    3.33    |    5.83    | **12.50**  | **15.00**  |  **18.33**  |
> > > | MATH500 L5 | Random (1P1S)                      |   29.11    | **41.05**  |   51.31    |   60.45    |    67.73    |
> > > |            | Strong Baseline (Unfiltered  Data) | **29.29**  |   39.74    |   51.31    |   59.33    |    66.98    |
> > > |            | Ours (RPD 1P3S)                    |   28.55    |   40.30    | **51.49**  | **61.20**  |  **69.97**  |
> > > | Olympiad   | Random (1P1S)                      |   19.14    |   27.45    |   35.68    |   45.48    |    52.89    |
> > > |            | Strong Baseline (Unfiltered  Data) |   20.33    |   29.75    |   38.95    | **47.26**  |    53.93    |
> > > |            | Ours (RPD 1P3S)                    | **20.40**  | **30.19**  | **39.10**  |   47.18    |  **54.16**  |
> > > | AMC23      | Random (1P1S)                      |   30.63    |   41.25    |   51.25    |   62.50    |    75.63    |
> > > |            | Strong Baseline (Unfiltered  Data) | **35.63**  |   44.38    | **56.88**  |   65.00    |    73.13    |
> > > |            | Ours (RPD 1P3S)                    |   34.38    | **46.25**  | **56.88**  | **68.13**  |  **80.00**  |

---

> > > > ### Author Response · Authors · 2025-11-20
> > > >
> > > > **W5 & Q1: Comparison with the strong baseline (unfiltered data).**
> > > >
> > > > * **Adaptive diversity strategy.** Furthermore, as shown in Table 8, the results align with the findings in our main paper. For moderately-solved problems (2-12 correct solutions), our method exhibits the highest diversity across both metrics; conversely, for well-solved problems (13-16 correct solutions), it shows lower diversity, indicating convergence. This confirms that our RPD-curated training equips the model with an adaptive strategy: increasing exploration on challenging problems while exploiting confident paths on simpler ones.
> > > >
> > > > **Table 8: Output Diversity Analysis (Qwen3-4B-Base)**
> > > >
> > > > | Method                     | Div-Self-BLEU (Pass 2-12) | Div-Self-BLEU (Pass 13-16) | Our Metric (Pass 2-12) | Our Metric (Pass 13-16) |
> > > > | :------------------------- | :-----------------------: | :------------------------: | :--------------------: | :---------------------: |
> > > > | Random (1P1S)              |           35.27           |           15.26            |         15.17          |          13.30          |
> > > > | Strong Baseline (Unfiltered Data) |           35.13           |           14.93            |         15.10          |          13.26          |
> > > > | Ours (RPD 1P3S)            |           38.20           |           14.31            |         15.80          |          12.62          |
> > > >
> > > > ---
> > > >
> > > > **W6: Improvements in subsequent RL training.**
> > > >
> > > > Thanks for raising this interesting question about subsequent RL training. To evaluate whether the diverse reasoning trajectories encouraged by our 1PNS paradigm yield a better foundation for subsequent reinforcement learning, we applied an additional RL phase on both the "Random (1P1S)" and "Ours (RPD 1P3S)" SFT checkpoints.
> > > >
> > > > * **Experimental setup.** We utilized the SimpleRL-Zoo dataset [1] and trained using GRPO within the veRL framework. Due to computational resource constraints, both models were trained for only 200 gradient update steps (Hyperparameters: LR $2\times10^{-5}$, KL coefficient 0.001, LoRA rank 32, rollout samples n=4 with temperature 0.7).
> > > > * **Superior RL performance.** As shown in Table 9, the model initialized with our RPD-curated 1P3S data consistently outperforms the Random 1P1S baseline. This confirms that the increased reasoning diversity from SFT provides a more robust initialization for policy optimization.
> > > >
> > > > **Table 9: Performance after RL (Qwen3-4B-Base)**
> > > >
> > > > | Benchmark    | SFT Method           | pass@1 (%) | pass@2 (%) | pass@4 (%) | pass@8 (%) | pass@16 (%) |
> > > > | :----------- | :------------------- | :--------: | :--------: | :--------: | :--------: | :---------: |
> > > > | **AIME24**   | Random (1P1S) + RL   |   10.00    |   16.67    |   19.17    |   24.17    |    32.50    |
> > > > |              | Ours (RPD 1P3S) + RL | **12.50**  | **17.50**  | **24.17**  | **30.84**  |  **37.50**  |
> > > > | **Olympiad** | Random (1P1S) + RL   |   41.77    |   50.23    |   57.27    |   62.84    |    67.66    |
> > > > |              | Ours (RPD 1P3S) + RL | **44.07**  | **52.67**  | **58.61**  | **64.69**  |  **68.99**  |
> > > > | **AIME25**   | Random (1P1S) + RL   |   10.83    |   15.83    |   20.84    |   25.83    |    34.17    |
> > > > |              | Ours (RPD 1P3S) + RL | **11.67**  | **20.00**  | **24.17**  | **30.00**  |  **36.67**  |
> > > > | **AMC23**    | Random (1P1S) + RL   |   47.50    |   60.00    |   70.00    |   78.75    |    80.63    |
> > > > |              | Ours (RPD 1P3S) + RL | **48.13**  | **61.25**  | **75.63**  | **81.88**  |  **84.38**  |
> > > >
> > > > ---
> > > >
> > > > **Q2: Choice of LoRA fine-tuning.**
> > > >
> > > > We selected QLoRA primarily to **save computational resources** and facilitate extensive experiments. Furthermore, recent research [2] indicates that this method **does not sacrifice model accuracy** compared to full-parameter fine-tuning when the configurations are appropriately set.
> > > >
> > > > ---
> > > >
> > > > [1] https://huggingface.co/collections/hkust-nlp/simplerl-zoo
> > > >
> > > > [2] LoRA Without Regret

---

> ### Author Response · Authors · 2025-11-27
>
> Dear Reviewer,
>
> I hope this message finds you well. As the discussion period is ongoing with one week remaining, I wanted to ensure we have addressed all your concerns satisfactorily. If there are any further points or feedback you'd like us to consider, please let us know. Your insights are invaluable to us, and we are eager to address any remaining issues to improve our work.
>
> Thank you for your time and effort in reviewing our paper.

---

### Official Review · Reviewer_Wzxw · 2025-10-24

**Soundness:** 2
**Presentation:** 3
**Contribution:** 2
**Rating:** 4
**Confidence:** 4

**Summary:**

The paper proposes a technique for selecting the SFT training dataset for better output diversity. In the context of LLM reasoning with math, the paper first posits that the current models suffer from a diversity issue. Then, it introduces a two-stage process that selects a subset of SFT data. They show that after tuning on this SFT data, the generated solutions on the test set achieve high accuracy under majority voting.

**Strengths:**

* clear writing
* identifies an important problem
* method description + experiment execution is sound

**Weaknesses:**

* I don't think diversity is a serious issue for math&code reasoning problems. They tend to be a problem for more subjective tasks. The problem domain selected by the author seems contrived --  i.e. since we have readily available benchmarks and datasets in math, let's do math
* There should be a temperature scaling for the majority vote. It's unclear why the authors stop at T=1 (Table 6). Also, even given the results in Table 6, it's clear that the marginal benefit of RPD diminishes as temperatures rise. Even DS-R1 (https://api-docs.deepseek.com/quick_start/parameter_settings) suggests production temperature up to 1.5. I think the author should sweep T much higher.
* The experiment setup is very contributed. OpenThought is a distillation dataset. Nobody uses it in actual production. The real question regarding diversity is how we should preserve output diversity during RLVR? The paper identifies the right question, but provides a solution in a setting (SFT) where the question is nonexistent.
* Even in the distillation setting, a natural baseline is the following: for each question in OpenThought, distill multiple solutions at a higher temperature, and then SFT on the new synthetic data.

**Questions:**

See weakness

---

> ### Author Response · Authors · 2025-11-20
>
> Thank you for your time and effort in reviewing our work. Below we respond to the comments in **Weaknesses (W)** and **Questions (Q)**.
>
> ---
>
> **W1: Importance of diversity in math reasoning.**
>
> We respectfully disagree with the reviewer's assertion that diversity is "not a serious issue for math and code reasoning problems". We believe this perspective may overlook the inherent attributes of the mathematical domain.
>
> **1. Diversity is an inherent property of the mathematical domain.**
> While a problem typically yields a single correct *answer*, it objectively admits multiple valid *reasoning paths* (e.g., algebraic vs. geometric, or recursive vs. combinatorial approaches). A model's failure to capture this multi-path reality (i.e., output homogenization) is not a sign of robustness; rather, it indicates **overfitting** to a single "canonical" path, which limits generalization.
>
> **2. Diversity is important.**
> This inherent diversity is not merely theoretical but has two important practical implications:
>
> * **Foundation for Test-Time Scaling (TTS):** The effectiveness of techniques like Best-of-N sampling and self-consistency *fundamentally depends* on the diversity of candidate solutions. If a model is "stuck" on a single path (i.e., mode collapse), sampling provides limited benefits [1-3].
> * **Initialization for Reinforcement Learning (RL):** The SFT model serves as the initial policy for subsequent RL [4]. If this policy is already "collapsed" to a single solution type, the RL agent is handicapped, as its exploration is confined to a narrow local optimum. Therefore, a diverse SFT model is an essential initialization for effective RL [1].
>
> ---
>
> **W2: Temperature scaling and marginal benefits.**
>
> Thank you for the detailed analysis regarding temperature settings. We would like to clarify that T=1.0 is already considered a high temperature for reasoning tasks (notably, T=0.6 is the recommended optimal temperature for the Qwen model family [5]).
>
> To directly address the reviewer's concern, we extended our experiments on the Qwen3-4B-Base model (MATH500 Level 5) to include higher temperatures (T=1.2 and T=1.5), covering a full sweep from T=0.2 to T=1.5.
>
> **Table 1: Performance Comparison across Temperatures (Qwen3-4B-Base, MATH500 L5)**
>
> | Method         | Temp (T) | pass@1 (%) | pass@2 (%) | pass@4 (%) | pass@8 (%) | pass@16 (%) |
> | :------------- | :------: | :--------: | :--------: | :--------: | :--------: | :---------: |
> | Random (1P1S)  |   0.2    | **51.12**  | **58.96**  |   65.30    |   69.96    |    73.13    |
> | **Ours (RPD)** | **0.2**  |   50.00    |   57.46    | **66.79**  | **71.46**  |  **74.82**  |
> | Random (1P1S)  |   0.4    | **54.11**  | **60.26**  |   68.10    |   72.39    |    76.31    |
> | **Ours (RPD)** | **0.4**  |   47.95    |   60.08    | **69.03**  | **75.19**  |  **77.80**  |
> | Random (1P1S)  |   0.6    |   49.26    |   60.64    |   66.98    |   72.20    |    77.43    |
> | **Ours (RPD)** | **0.6**  | **52.61**  | **61.57**  | **71.64**  | **75.94**  |  **79.29**  |
> | Random (1P1S)  |   0.8    | **51.87**  | **61.57**  |   69.22    |   74.44    |    78.36    |
> | **Ours (RPD)** | **0.8**  |   50.56    |   60.82    | **69.47**  | **74.82**  |  **78.92**  |
> | Random (1P1S)  |   1.0    |   45.34    |   59.71    |   68.66    | **73.88**  |    76.87    |
> | **Ours (RPD)** | **1.0**  | **48.51**  | **59.89**  | **69.22**  | **73.88**  |  **77.80**  |
> | Random (1P1S)  |   1.2    |   32.84    |   45.34    |   56.72    |   65.68    |    71.83    |
> | **Ours (RPD)** | **1.2**  | **35.07**  | **50.00**  | **57.65**  | **66.98**  |  **73.88**  |
> | Random (1P1S)  |   1.5    |    2.99    |    4.85    |    8.40    |   14.00    |    22.02    |
> | **Ours (RPD)** | **1.5**  |  **5.22**  |  **6.91**  | **13.99**  | **18.84**  |  **27.05**  |
>
> **1. Performance degrades sharply at high temperatures.**
> As the results demonstrate, performance for both methods drops significantly after T=1.0, confirming that settings above this threshold are generally impractical for reasoning task.
>
> **2. The performance gap reflects a Quality-Diversity inflection point.**
> We acknowledge that the advantage of RPD narrows at intermediate temperatures T=0.8. However, the gap widens again as the temperature increases further (from T=1.0 to T=1.5). This implies an inflection point near T=0.8, where the baseline temporarily strikes a balance—gaining sampling diversity without suffering severe quality collapse.
>
> **3. RPD remains strictly superior in practical settings.**
> Crucially, our method remains superior to the baseline across all tested temperatures, particularly at larger values of k in pass@k. Most significantly, our peak performance gain occurs at **T=0.6** (the recommended optimal setting), validating the practical value of our approach in realistic deployment.

---

> > ### Author Response · Authors · 2025-11-20
> >
> > **W3: About OpenThought dataset and the Necessity of SFT Diversity.**
> >
> > **1. OpenThought3 is a valid resource for reasoning research.**
> > We respectfully disagree with the claim that OpenThought dataset is not used in production. To our best knowledge, OpenThought has been utilized by recent works for training reasoning models [6,7].
> >
> > **2. Diversity in SFT is the essential foundation for RL.**
> > Regarding the second point, while we agree that diversity in the RL phase is a critical research question, we argue that this **reinforces**, rather than negates, the importance of diversity in the SFT stage. The SFT model is not an isolated step; it serves as the *initial policy* and starting point for any subsequent RL tuning. If this initial SFT policy is already "collapsed"—that is, it has overfit to a single, homogenous solution path—then the RL agent is severely handicapped before its training even begins. Its exploration will be confined to a very narrow, suboptimal local optimum [1,4]. Therefore, ensuring the SFT model has a rich and diverse understanding of the solution space is not solving a "nonexistent" problem. It is a crucial prerequisite for providing a more effective and robust foundation for the subsequent RL phase, enabling it to explore a much richer policy space.
> >
> > ---
> >
> > **W4: Suggested baseline of high-temperature data generation.**
> >
> > Thank you for your suggestion. However, we believe this proposed baseline is not directly comparable, for two main reasons related to our paper's scope and the fundamental constraints of math reasoning tasks.
> >
> > **1. Scope of Contribution (Selection vs. Generation).**
> > Our paper's central research question is how to *select* an optimal, diverse subset from a *given, fixed* dataset. Our method (RPD) is a **curation strategy**, not a generation strategy. The suggestion—to generate an entirely new dataset by changing the distillation process—is a different (though valid) research problem, but it does not serve as a direct baseline for evaluating our selection algorithm.
> >
> > **2. The Primacy of Quality in Mathematical Reasoning.**
> > For a domain like mathematics, logical coherence and correctness are paramount. A dataset is truly valuable only if it maintains high quality.
> >
> > * The OpenThought3 dataset (our source pool) was generated using QwQ-32B at T=0.7. This is *already* higher than the recommended optimal temperature (T=0.6) for that model (https://huggingface.co/Qwen/QwQ-32B), which was chosen to balance quality and diversity.
> > * In our own preliminary tests, pushing the temperature much higher (e.g., T=1.0 - 1.5) to "increase diversity" does not yield a usable dataset; it results in a sharp degradation of quality, producing outputs with significant incoherence.
> > * Identifying the optimal temperature to balance the Quality-Diversity trade-offs is a separate, complex generation-focused task. From our perspective as a *curation* method, a new dataset with higher diversity is not a necessary or appropriate baseline.
> >
> > ---
> >
> > [1] Preserving Diversity in Supervised Fine-Tuning of Large Language Models
> >
> > [2] Inference-Aware Fine-Tuning for Best-of-N Sampling in Large Language Models
> >
> > [3] Rethinking Fine-Tuning when Scaling Test-Time Compute: Limiting Confidence Improves Mathematical Reasoning
> >
> > [4] SFT Memorizes, RL Generalizes: A Comparative Study of Foundation Model Post-training
> >
> > [5] Qwen3 Technical Report
> >
> > [6] LoRA Without Regret
> >
> > [7] Reverse-Engineered Reasoning for Open-Ended Generation

---

> ### Author Response · Authors · 2025-11-27
>
> Dear Reviewer,
>
> I hope this message finds you well. As the discussion period is ongoing with one week remaining, I wanted to ensure we have addressed all your concerns satisfactorily. If there are any further points or feedback you'd like us to consider, please let us know. Your insights are invaluable to us, and we are eager to address any remaining issues to improve our work.
>
> Thank you for your time and effort in reviewing our paper.

---

### Official Review · Reviewer_YtMr · 2025-10-31

**Soundness:** 3
**Presentation:** 3
**Contribution:** 2
**Rating:** 4
**Confidence:** 3

**Summary:**

This paper introduces Reasoning Path Divergence (RPD), a step-level semantic distance metric designed to quantify diversity among long chain-of-thought (CoT) reasoning paths in LLMs. Using RPD, the authors curate a one-problem-multiple-solutions (1PNS) training set from the OpenThought3 dataset and fine-tune a 4B-parameter model (Qwen3-Base). The proposed method improves pass@16 performance by 2.80% on average across three math benchmarks, with a peak gain of 4.99% on AIME24. The paper argues that training with diverse reasoning paths mitigates output homogenization and enhances test-time scaling (TTS).

**Strengths:**

### 1. Novel and Well-Motivated Metric (RPD):

RPD is a creative and principled approach to measuring semantic diversity at the step level, addressing a key limitation of embedding-based methods that conflate surface-level differences with strategic divergence. The asymmetric design is particularly insightful for handling summarization granularity.


###  2. Thorough Experimental Design:

The paper includes extensive ablations, scalability tests, and diversity analyses. The authors also validate their LLM judge against human annotations (78% accuracy), lending credibility to their automated evaluation pipeline.

**Weaknesses:**

### 1. Limited Generalization Beyond Math

All experiments are conducted on math reasoning tasks (AIME24, MATH500, Olympiad Bench). While the gains are convincing, it is unclear whether RPD and 1PNS generalize to other reasoning domains (e.g., logic, science, coding), limiting the broader impact of the work.

### 2. Scalability and Compute Overhead

RPD relies on LLM-based summarization and embedding computation for every solution pair, which is compute-intensive and may not scale well to larger datasets or real-time curation. The paper does not discuss the computational cost of RPD compared to simpler baselines (e.g., raw embeddings).

###  3. Lack of Theoretical Justification

While RPD is intuitively appealing, the paper does not provide theoretical grounding for why step-level asymmetric distance correlates with strategic diversity or why 1PNS improves TTS. The connection between training data diversity and inference-time exploration remains largely empirical.

**Questions:**

see weakness.

---

> ### Author Response · Authors · 2025-11-20
>
> Thank you for your supportive review and suggestions. Below we respond to the comments in **Weaknesses (W)** and **Questions (Q)**.
>
> ---
>
> **W1: Limited Generalization Beyond Math.**
>
> Thank you for your suggestion to test RPD's effectiveness beyond the math domain. We have conducted new experiments extending our pipeline to the **code generation domain**.
>
> * **New Experiment Setup:** We curated a 1P3S RPD dataset and a 1P1S Random baseline dataset containing 300 code training samples from OpenThought3. We mixed this with our original math dataset and fine-tuned the Qwen3-4B-Base model (10 epochs).
> * **Evaluation:** We tested on both the original math benchmarks and two new code benchmarks: **Live Code Bench** and **HumanEval**.
> * **Results:** As shown in the table below, our method **consistently outperforms the baselines across all evaluated datasets in both math and code domains**. This universal improvement confirms that RPD and the 1PNS paradigm are generalizable principles for enhancing reasoning diversity.
>
> | Benchmark           | Method                    |  pass@1   |  pass@2   |  pass@4   |  pass@8   |  pass@16  |
> | :------------------ | :------------------------ | :-------: | :-------: | :-------: | :-------: | :-------: |
> | **Math Benchmarks** |                           |           |           |           |           |           |
> | AIME24              | Base Model                |   8.34    |   13.33   |   16.67   |   21.67   |   27.50   |
> |                     | Random 1P1S (Math + Code) |   10.84   |   15.00   |   19.17   |   23.33   |   29.17   |
> |                     | **Ours (Math + Code)**    | **12.50** | **16.67** | **22.50** | **26.67** | **31.67** |
> | MATH500 L5          | Base Model                |   46.08   |   56.90   |   64.37   |   71.27   |   75.00   |
> |                     | Random 1P1S (Math + Code) |   45.71   |   57.46   |   64.56   |   69.03   |   75.93   |
> |                     | **Ours (Math + Code)**    | **48.13** | **57.84** | **67.91** | **72.57** | **77.24** |
> | Olympiad Bench      | Base Model                |   39.54   |   47.11   |   53.56   |   61.13   |   65.95   |
> |                     | Random 1P1S (Math + Code) |   39.84   |   47.92   |   53.94   |   60.61   |   66.18   |
> |                     | **Ours (Math + Code)**    | **39.99** | **49.04** | **55.20** | **61.72** | **67.29** |
> | **Code Benchmarks** |                           |           |           |           |           |           |
> | Live Code Bench     | Base Model                |   13.46   |   22.46   |   30.05   |   35.35   |   38.86   |
> |                     | Random 1P1S (Math + Code) |   14.22   |   21.99   |   30.05   |   36.30   |   40.61   |
> |                     | **Ours (Math + Code)**    | **17.35** | **25.88** | **32.04** | **38.10** | **42.56** |
> | HumanEval           | Base Model                |   2.86    |   5.36    |   9.56    |   15.93   |   25.00   |
> |                     | Random 1P1S (Math + Code) |   18.64   |   33.54   |   47.26   |   66.36   |   80.73   |
> |                     | **Ours (Math + Code)**    | **23.36** | **38.74** | **57.73** | **75.44** | **87.20** |
>
> ---
>
> **W2: Scalability and Compute Overhead.**
>
> We have profiled our pipeline on 2x H20 GPUs and found the computational overhead to be highly acceptable for dataset construction.
>
> * **Curation Breakdown (1.68s/solution):** The total time to curate one solution comprises LLM Summarization (1.64s using Qwen3-14B on the OpenThought dataset) and Embedding (0.04s). The final pairwise distance calculation is negligible (5.6ms/problem).
> * **Trade-off vs. Raw Baseline:** We acknowledge that our RPD pipeline (1.68s) entails slightly more overhead than the Raw Embedding baseline (0.91s). However, we consider this modest additional time (<1s) a worthy trade-off given the superior performance achieved by our method.
> * **Acceptable Overhead Relative to Generation & Reuse:** The 1.68s curation cost is significantly faster than the time required to generate a single Long CoT solution (approx. 4.04s on a 4B model), and this fraction shrinks further when using larger models. Moreover, RPD is strictly a **one-time construction cost**. Once the high-quality dataset is built, it can be extensively reused by the community for repeated training. Compared to the massive cumulative compute of these numerous training runs, our initial single-pass curation cost is minimal.
>
> | Curation Step  | Method                             | Time (per solution) |
> | :------------- | :--------------------------------- | :------------------ |
> | **Ours (RPD)** | LLM Summarization (Qwen3-14B)      | 1.64s               |
> |                | Embedding (Qwen3-Embedding-8B)     | 0.04s               |
> |                | **Total Cost**                     | **1.68s**           |
> | **Baseline**   | Raw Embedding (Qwen3-Embedding-8B) | 0.91s               |

---

> > ### Author Response · Authors · 2025-11-20
> >
> > **W3: Lack of Theoretical Justification.**
> >
> > We acknowledge that our work is empirically driven; however, our approach is grounded in prior theoretical research regarding the limitations of standard SFT.
> >
> > * **Theoretical Foundation ("Diversity Collapse"):** The standard 1P1S SFT, using Cross-Entropy (CE) loss, is known to cause "overconfidence" and "diversity collapse" [1, 2]. The CE mechanism indiscriminately penalizes all non-target tokens to maximize the likelihood of the single target path provided in the data. This process acts as an "unbounded probability transfer", driving the probabilities of valid latent alternatives toward zero. Consequently, this "distribution collapse" fundamentally reduces the output diversity required for effective Test-Time Scaling [1,2].
> > * **Data-Centric Solution (1PNS):** Our 1PNS paradigm acts as a direct countermeasure to this collapse. By maximizing likelihood over multiple disjoint yet valid reasoning paths ($y_1, y_2...$) for the same input $x$, we force the model to learn a **multi-peaked posterior distribution** rather than a sharp, unimodal one. This training-time multi-modality directly preserves the entropy required for inference-time sampling, allowing the model to access distinct reasoning "peaks" that are otherwise suppressed in the collapsed 1P1S distribution.
> > * **RPD Correlates with Strategic Diversity:** RPD structurally measures "Logical Inclusion" via asymmetric step-matching, identifying unique reasoning steps that *cannot be mapped* to the reference solution. This ensures high scores strictly reflect strategic novelty rather than mere syntactic noise.
> >
> > Therefore, while our contribution is an empirical methodology, it is built directly upon the established theoretical understanding.
> >
> > ---
> >
> > We hope these new experiments and analyses fully address your concerns. We will incorporate these detailed discussions into the final version of the paper.
> >
> > ---
> >
> > [1]Preserving Diversity in Supervised Fine-Tuning of Large Language Models
> >
> > [2]Rethinking Fine-Tuning when Scaling Test-Time Compute: Limiting Confidence Improves Mathematical Reasoning

---

> ### Author Response · Authors · 2025-11-27
>
> Dear Reviewer,
>
> I hope this message finds you well. As the discussion period is ongoing with one week remaining, I wanted to ensure we have addressed all your concerns satisfactorily. If there are any further points or feedback you'd like us to consider, please let us know. Your insights are invaluable to us, and we are eager to address any remaining issues to improve our work.
>
> Thank you for your time and effort in reviewing our paper.

---

### Official Review · Reviewer_d6L6 · 2025-11-03

**Soundness:** 3
**Presentation:** 3
**Contribution:** 3
**Rating:** 6
**Confidence:** 3

**Summary:**

This paper addresses low output diversity by introducing Reasoning Path Divergence (RPD), a step-level metric for quantifying semantic diversity between reasoning paths. Using RPD, the authors curate a one-problem–multiple-solutions (1PNS) training dataset that maximizes strategic variation across Long-CoT solutions. Fine-tuning a Qwen3-4B-Base model on this RPD-curated dataset yields consistent gains over standard one-problem–one-solution (1P1S) baselines. The method increases reasoning diversity without requiring architectural or inference-time changes.

**Strengths:**

- The metric design is clear and reasonable. RPD introduces a fine-grained, asymmetric step-level comparison that captures strategic rather than superficial differences.
- Consistent improvements across AIME24, MATH500, and OlympiadBench, with robust ablations on number of solutions, problem selection, and temperature scaling.

**Weaknesses:**

- Evaluation limited to math reasoning: The study focuses exclusively on quantitative tasks (AIME, MATH, Olympiad); generalization to open-ended or commonsense reasoning remains unclear.

- The curation pipeline—step summarization, embedding, and pairwise distance computation—may be costly for larger datasets.

**Questions:**

n/a

---

> ### Author Response · Authors · 2025-11-20
>
> Thanks for your time and effort in reviewing our work, as well as for recognizing our contributions! Below we respond to the comments in **Weaknesses (W)** and **Questions (Q)**.
>
> ---
>
> **W1: Concern about evaluation limited to math reasoning (Generalization to OOD)**
>
> Thanks for your suggestion. We conduct new experiments on two out-of-domain (OOD) benchmarks: TruthfulQA and MMLU-Pro (Biology & CS subsets). We compare Ours (RPD 1PNS) against the Random 1P1S baseline on Qwen2.5-3B and Qwen3-4B-Base.
>
> * **Consistent Superiority on OOD Tasks.** As shown in the tables below, RPD 1PNS consistently outperforms Random 1P1S across all tested OOD benchmarks (TruthfulQA and MMLU-Pro). This demonstrates that the reasoning diversity captured by our method translates effectively to unseen domains.
>
> **Table 1: OOD benchmark results for Qwen2.5-3B**
>
> | Benchmark          | Method              | pass@1    | pass@2    | pass@4    | pass@8    | pass@16   |
> | :----------------- | :------------------ | :-------- | :-------- | :-------- | :-------- | :-------- |
> | **TruthfulQA**     | Random 1P1S         | **63.16** | 74.05     | 80.89     | 87.59     | 91.39     |
> |                    | **Ours (RPD 1PNS)** | 62.15     | **74.68** | **83.42** | **90.25** | **93.67** |
> | **MMLU-Pro (Bio)** | Random 1P1S         | 28.59     | 43.93     | 60.11     | 76.15     | 85.08     |
> |                    | **Ours (RPD 1PNS)** | **33.47** | **45.33** | **67.92** | **77.96** | **85.50** |
> | **MMLU-Pro (CS)**  | Random 1P1S         | 23.66     | **33.66** | 49.76     | 65.61     | 80.24     |
> |                    | **Ours (RPD 1PNS)** | **24.63** | 30.49     | **50.49** | **66.10** | **82.20** |
>
> **Table 2: OOD benchmark results for Qwen3-4B-Base**
> | Benchmark          | Method              | pass@1    | pass@2    | pass@4    | pass@8    | pass@16   |
> | :----------------- | :------------------ | :-------- | :-------- | :-------- | :-------- | :-------- |
> | **TruthfulQA**     | Random 1P1S         | 64.30     | 73.54     | 81.14     | 85.19     | 87.22     |
> |                    | **Ours (RPD 1PNS)** | **64.94** | **74.30** | **82.15** | **86.08** | **88.99** |
> | **MMLU-Pro (Bio)** | Random 1P1S         | 73.78     | 82.57     | 88.15     | 91.35     | 94.56     |
> |                    | **Ours (RPD 1PNS)** | **74.06** | **83.54** | **88.28** | **92.05** | **94.98** |
> | **MMLU-Pro (CS)**  | Random 1P1S         | **58.05** | 69.51     | 78.78     | 86.34     | 90.73     |
> |                    | **Ours (RPD 1PNS)** | 56.10     | **71.22** | **80.00** | **87.56** | **91.46** |
>
> ---
>
> **W2: Concern about the computational cost of the curation pipeline**
>
> We have profiled our pipeline on 2x H20 GPUs and found the computational overhead to be highly acceptable for dataset construction.
>
> * **Curation Overhead (1.68s/solution).** The total time to curate one solution is composed of LLM Summarization (1.64s, using Qwen3-14B on the OpenThought dataset) and Embedding (0.04s). The final pairwise distance calculation is negligible (5.6ms/problem).
> * **Faster than Generation.** This 1.68s curation cost is significantly faster than the time required to generate a single Long CoT solution (approx. 4.04s on a 4B model). In practical settings where data is generated by larger models (e.g., 32B+), our curation overhead becomes an even smaller fraction of the total pipeline.
> * **One-Time Cost for Extensive Reuse.** RPD is strictly a one-time construction cost. Given that our method produces a high-quality dataset, it can be extensively reused by the community to train models repeatedly. Compared to the massive cumulative compute of these numerous training runs, our initial single-pass curation cost is virtually negligible.
>
> ---
>
> We hope these new experiments and cost analyses fully address your concerns. We will incorporate these detailed discussions into the final version of the paper.

---

> ### Author Response · Authors · 2025-11-27
>
> Dear Reviewer,
>
> I hope this message finds you well. As the discussion period is ongoing with one week remaining, I wanted to ensure we have addressed all your concerns satisfactorily. If there are any further points or feedback you'd like us to consider, please let us know. Your insights are invaluable to us, and we are eager to address any remaining issues to improve our work.
>
> Thank you for your time and effort in reviewing our paper.

---

### Note · Authors · 2026-01-28

I have read and agree with the venue's withdrawal policy on behalf of myself and my co-authors.

---

### Meta-Review · Area_Chair_rGnN · 2026-01-06

**Summary:**

This paper proposes Reasoning Path Divergence (RPD), a step-level semantic distance metric designed to quantify diversity among chain-of-thought (CoT) reasoning paths. Based on this metric, the authors introduce a curation strategy to construct one-problem-multiple-solutions (1PNS) datasets for Supervised Fine-Tuning (SFT). The authors claim that fine-tuning on RPD-curated data improves performance on mathematical reasoning benchmarks by mitigating mode collapse and enhancing reasoning diversity.

**Reviewer Concerns:**

The initial reviews were mixed-to-negative (6, 4, 4, 4), with reviewers raising significant concerns regarding the experimental scale, limited generalization beyond math, computational overhead, and the choice of baselines. The authors provided a comprehensive rebuttal containing substantial new experiments.

**Concerns Addressed by Rebuttal:**
1.  **Generalization:** The authors added experiments on code generation (LiveCodeBench, HumanEval) and OOD benchmarks (TruthfulQA, MMLU-Pro) to address concerns from Reviewers d6L6 and YtMr.
2.  **Metric Stability:** In response to Reviewer xNim, the authors demonstrated that RPD calculation remains robust when using a smaller 7B model for summarization.
3.  **Model Architecture:** Validation was extended to Llama-3.1-8B-Instruct to show independence from the Qwen family.
4.  **Computational Cost:** A breakdown of the curation overhead was provided to address efficiency concerns.

**Outstanding Concerns:**
1.  **Experimental Scale:** The primary results rely on a very small dataset (300 samples). While the authors provided a 3,000-sample extension during the rebuttal, this was trained for only 1 epoch due to compute constraints. This scale is still significantly smaller than standard instruction-tuning datasets. Consequently, the practical utility and robustness of the method for production-scale model training remain unproven.
2.  **Baseline Appropriateness:** Reviewer Wzxw suggested comparing the selection method against a generative baseline (distilling multiple solutions at higher temperatures to create new data). The authors dismissed this as ``out of scope'' for a curation paper. However, from a practical standpoint, if simple high-temperature generation yields better results than complex RPD-based selection, the utility of the proposed method is questionable. The lack of this direct comparison is a significant omission.
3.  **Scope of Revisions:** The rebuttal introduced entirely new domains, model architectures, benchmarks, and a preliminary RL experiment. The necessity of such extensive additions suggests that the original submission was effectively incomplete. These new results constitute a major revision that requires thorough peer review, which is not possible within the rebuttal window, especially given the lack of reviewer engagement with the new data.

**Reviewer Scores:**

-   **Reviewer d6L6 (6):** Likely to change to 4. The reviewer found the work "marginally above" but explicitly stated they "would not mind if paper is rejected".
-   **Reviewer YtMr (4):** Likely to remain at 4. While the generalization concerns were addressed, the fundamental issues regarding scale and theoretical justification likely prevent a full shift to acceptance.
-   **Reviewer Wzxw (4):** Likely to remain at 4. The reviewer's core critique regarding the necessity of the method versus simple generative baselines was argued against rather than empirically resolved.
-   **Reviewer xNim (4):** Likely to remain at 4. The reviewer would likely appreciate the extensive new experiments (RL, Llama) but retain reservations about the complexity of the pipeline versus the marginal gains over simpler methods at small scales.

---

### Decision · Program_Chairs · 2026-01-26

Reject